# Heart rate and heart rate variability in horses undergoing hot and cold shoeing

**Onjira Huangsaksri**[1,2]**, Thita Wonghanchao**[1,2]**, Kanokpan Sanigavatee**[1,2]**, Chanoknun Poochipakorn**[1,2]**, Metha Chanda**[2,3]*

**1** Veterinary Clinical Study Programme, Graduate School, Kasetsart University, Kamphaeng Saen Campus, Nakorn Pathom, Thailand, **2** Faculty of Veterinary Medicine, Department of Large Animal and Wildlife Clinical Science, Kasetsart University, Kamphaeng Saen Campus, Nakorn Pathom, Thailand, **3** Thailand Equestrian Federation, Sports Authority of Thailand, Bangkok, Thailand

* fvetmtcd@ku.ac.th

**Data Availability Statement:** The HRV data supporting this study's findings are available at https://doi.org/10.6084/m9.figshare.2465499.v1.

## Abstract

Heart rate variability (HRV) is a frequently used indicator of autonomic responses to various stimuli in horses. This study aimed to investigate HRV variables in horses undergoing cold ($n = 25$) or hot ($n = 26$) shoeing. Multiple HRV variables were measured and compared between horses undergoing cold and hot shoeing, including the time domain, frequency domain, and nonlinear variables pre-shoeing, during shoeing, and at 30-minute intervals for 120 minutes post-shoeing. The shoeing method interacted with time to change the HRV variables standard deviation of RR intervals (SDNN), root mean square of successive RR interval differences (RMSSD), very-low-frequency band, low-frequency band (LF), the LF to high-frequency band ratio, respiratory rate, total power, standard deviation perpendicular to the line of identity (SD1), and standard deviation along the line of identity (SD2). SDNN, RMSSD, and total power only increased 30 minutes after hot shoeing (all $p < 0.05$). Triangular interpolation of normal-to-normal intervals (TINN) and the HRV triangular index increased during and up to 120 minutes after hot shoeing ($p < 0.05$–$0.001$). TINN increased only during cold shoeing ($p < 0.05$). LF increased 30 and 60 minutes after hot shoeing (both $p < 0.05$). SD1 and SD2 also increased 30 minutes after hot shoeing (both $p < 0.05$). SDNN, TINN, HRV triangular index, LF, total power, and SD2 were higher in hot-shoed than cold-shoed horses throughout the 120 minutes post-shoeing. Differences in HRV were found, indicating increased sympathovagal activity in hot shoed horses compared to cold shoed horses.

## Introduction

Horseshoeing is a mechanical method in which a steel bar is fitted to the hoof's solar aspect. It promotes healthy hoof function and facilitates coping with loading stress on the foot, reducing the risk of lameness [1,2]. It is also used to prevent the splitting of the hoof wall, slipping [1], and bruising [3]. Cold and hot shoeing methods are widely used in horses [4,5] to enhance performance and prolong career longevity [6]. Subjectively, horses should benefit more from

**Funding:** This work was partially supported by the Kasetsart Veterinary Development Funds. The funders had no role in study design, data collection and analysis, decision to publish, or manuscript preparation.

**Competing interests:** The authors have declared that no competing interests exist.

hot than cold shoeing since it improves the juxtaposition between the hoof's ground contact surface and the metal shoe [1]. Furthermore, horseshoes can be fitted on hooves more accurately with the hot shoeing method [7]. Nonetheless, the lack of an objective evaluation makes it unclear whether horses benefit more from hot than cold shoeing.

Various studies have investigated heart rate variability (HRV) in horses in response to different psychological and physiological challenges [8–10]. Therefore, the changes in HRV variables have been adopted to indicate autonomic responses to hot and cold shoeing protocols. HRV refers to the time difference between beat-to-beat (RR) intervals during the cardiac cycle [9,11]. The fluctuation in time intervals reflects the oscillatory influences of the parasympathetic (vagal) nervous system (PNS) and sympathetic nervous system (SNS) acting on the heart's sinoatrial node [12–14]. This variation in the cardiac cycle helps maintain cardiovascular homeostasis and automatic responses to challenges [15,16]. HRV also has the advantage of being a non-invasive biomarker that indicates the physiological stress responses of the autonomic nervous system (ANS) in horses under several conditions, including clinical examination [17], exercise [18], road transportation [14,19], and cardiac disease [8,20].

HRV variables can be used to assess autonomic responses in horses using different methods, including time domain, frequency domain, and nonlinear result analyses. In general, sympathetic and vagal components modulate the RR intervals, the standard deviation of RR intervals (SDNN), and the low-frequency (LF) band [11,21]. In contrast, the effects of vagal activity on the sinus rhythm are reflected in changes in the root mean square of successive RR interval differences (RMSSD), the relative number of successive RR interval pairs that differ by >50 ms (pNN50), and the high-frequency (HF) band [9,21]. Heart rate (HR) and the LF/HF ratio were increased, while SDNN and RMSSD were decreased in horses with severe gastric ulcers, indicating reduced vagal activity [22]. Geometric analyses that measure the density distribution of RR intervals, such as the triangular interpolation of normal-to-normal intervals (TINN) and HRV triangular index, are used to estimate the overall variation of the heartbeat [11,21]. In addition, nonlinear analyses, consisting of the standard deviation perpendicular to the line of identity (SD1) and the standard deviation along the line of identity (SD2), are used to indicate short-term and long-term HRV, respectively [11]. The TINN, HRV triangular index, SD1, and SD2 decreased in exercising horses, indicating decreased HRV [23,24]. Therefore, HRV is a promising indicator for detecting the ANS's response to challenges in horses.

Given the scarcity of studies on HRV changes in horses after conventional shoeing methods and comparisons of the effects of different shoeing methods on horses' HRV variables, this study aimed to investigate the impact of conventional shoeing methods on HRV variables and compare these variables between horses undergoing hot and cold shoeing.

## Materials and methods

### Horses

Fifty-one healthy horses (22 geldings and 29 mares, aged 12–19 years) were recruited from different equestrian clubs in Thailand: 25 from the Thai Polo and Equestrian Club (12˚53'29.8 "N 100˚59'37.0" E), six from the House of Horse Riding Club (13˚43'27.6 "N 100˚40'54.6" E), two from the Checkmate Riding Club (13˚52'48.2 "N 100˚38'57.3" E), and 18 from the Horse Lover's Club (13˚59'38.2 "N 100˚40'53.9" E). The horses were housed in $3 \times 5$ m$^2$ stables with straw bedding and fed a similar daily diet consisting of 1–3 kg of commercial pellets and approximately 10 kg of hay per horse. Water was provided ad libitum in the stable.

The horses had been routinely trained and used for equestrian competitions or in riding schools. Each horse had undergone routine shoeing practice with familiar farriers and received either hot ($n = 26$) or cold ($n = 25$) shoeing (S1 Table) at 4–5-week intervals according to the

specified methods [1,25]. The inclusion criteria for horses in this study were as follows: (1) experience of either hot or cold shoeing protocols, (2) no visible lesions that could lead to lameness on the hoof structure, and (3) not undergoing lameness treatment or any other therapeutic protocol before the experiment. Any horses that suffered complications from farrier practice were excluded from this study. However, no horses experienced complications after both shoeing protocols. Therefore, the HRV analysis included all 51 horses recruited for this study. The animal experiments in this study were approved by Kasetsart University's Institute of Animal Care and Use Committee (ACKU65-VET-051).

## Experimental protocol

This study was designed to minimize psychological stress by (1) experimenting on horses accustomed to fitting with each type of farrier practice and (2) having familiar farriers shoe the horses in familiar places. These arrangements aimed to reduce confounding factors such as fear and anxiety with unfamiliarity, which can distort the HRV changes during shoeing. Before the shoeing protocol, the horses were equipped with a Polar equine HR monitor (HRM) set (Polar Electro Oy, Kempele, Finland) to record their RR intervals during the experiment. The HRM device has been proven to accurately measure HRV variables in horses [13,26,27]. Briefly, the Polar equine belt was soaked with water to augment electrical signal transmission and then attached to an HR sensor (Polar H10). Next, the belt set was fastened to the horse's chest, with the sensor in the middle of its left side (Figs 1A, 2A and 2B). Finally, the sensor was connected to the latest version of a sports watch used to measure HRV in human studies (Polar Vantage V2) [28].

After fitting the HRM device, the horses were given approximately 10 minutes to become familiar with the setup. Then, RR intervals were recorded within designated stables, where the condition was similar to housing stables, in a farrier place for 30 minutes before hot or cold shoeing on all four legs. The horses were shod by their familiar farriers at usual places. First, the old shoes were removed from the horse's feet, and the sole and frog were cleaned with a hoof knife. Next, the hoof was trimmed to remove the overgrowth on the hoof wall and adjust the hoof symmetry and angle before fitting the new shoe. Then, the hoof's solar surface was rasped to provide an even ground contact surface for the foot. When performing hot shoeing, a shoe of the appropriate size was heated in the forge and placed briefly onto the hoof's solar margin to sear the contact surface where it would finally be placed (Fig 1B and 1C). Then, the seared feet were fitted with the cooled shoes using conventional shoeing techniques. Cold shoeing was performed similarly, except the metal shoes were not heated before shoeing. The shoeing protocols took 45–60 minutes for each horse. The RR intervals were recorded from 30 minutes before shoeing began to 120 minutes afterward. Therefore, each horse's total RR recording period was approximately 195–210 minutes.

## Data acquisition

The raw datasets derived from the Polar sports watch were uploaded to the Polar flow program (https://flow.polar.com/) and then analyzed for the HRV variables using the Kubios Premium software (Kubios HRV Scientific; https://www.kubios.com/hrv-premium/). However, technical errors, mainly artifacts, may arise when using RR detectors [29,30]. This software includes an automatic artifact correction algorithm that is more accurate than the standard version and has been validated to correct artifacts and ectopic beats in inter-beat interval (IBI) data [31]. The Kubios HRV Scientific software also supports automatic noise detection to identify noise segments from the IBI data that distort various consecutive beat detections.

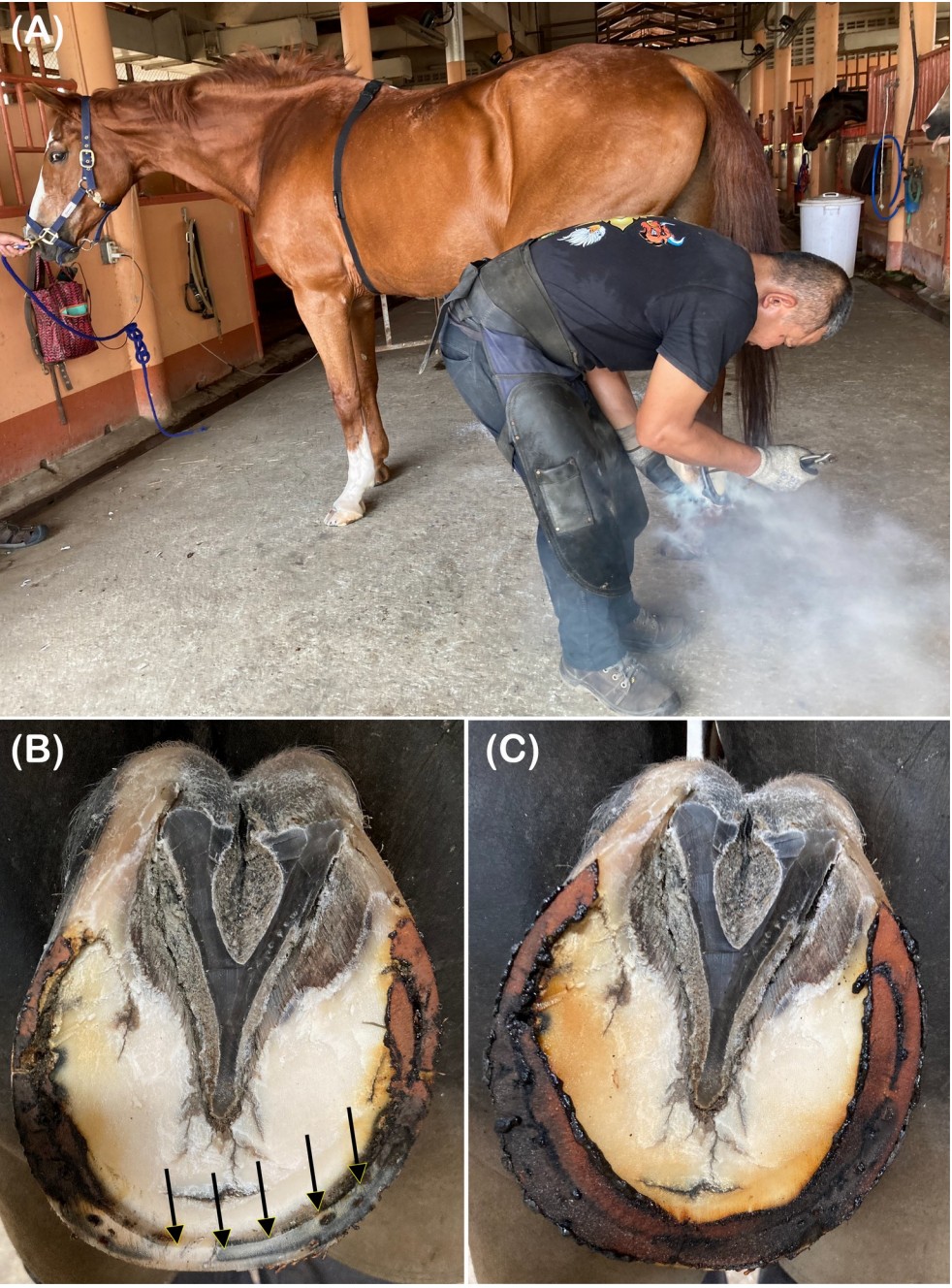

**Fig 1.** (A) A horse undergoes the hot shoeing protocol while equipped with the HRM device. (B) The solar surface of the hoof is partially seared, leaving an intact area at the toe (black arrows) after the first placing of the heated metal shoe on the ground contact surface. (C) The burned component is rasped out, and the hoof's solar surface is entirely seared after the second placing of the heated metal shoe.

Before analyzing HRV variables, smoothness priors were used to remove IBI time series non-stationarities. The cutoff frequency was set at 0.035 Hz per the user guideline (https://www.kubios.com/downloads/Kubios_HRV_Users_Guide.pdf). The autonomic correction was set to remove artifacts and ectopic beats present in the IBM data. The automatic noise detection was fixed at a medium level. The variables included (1) time domain variables (RR

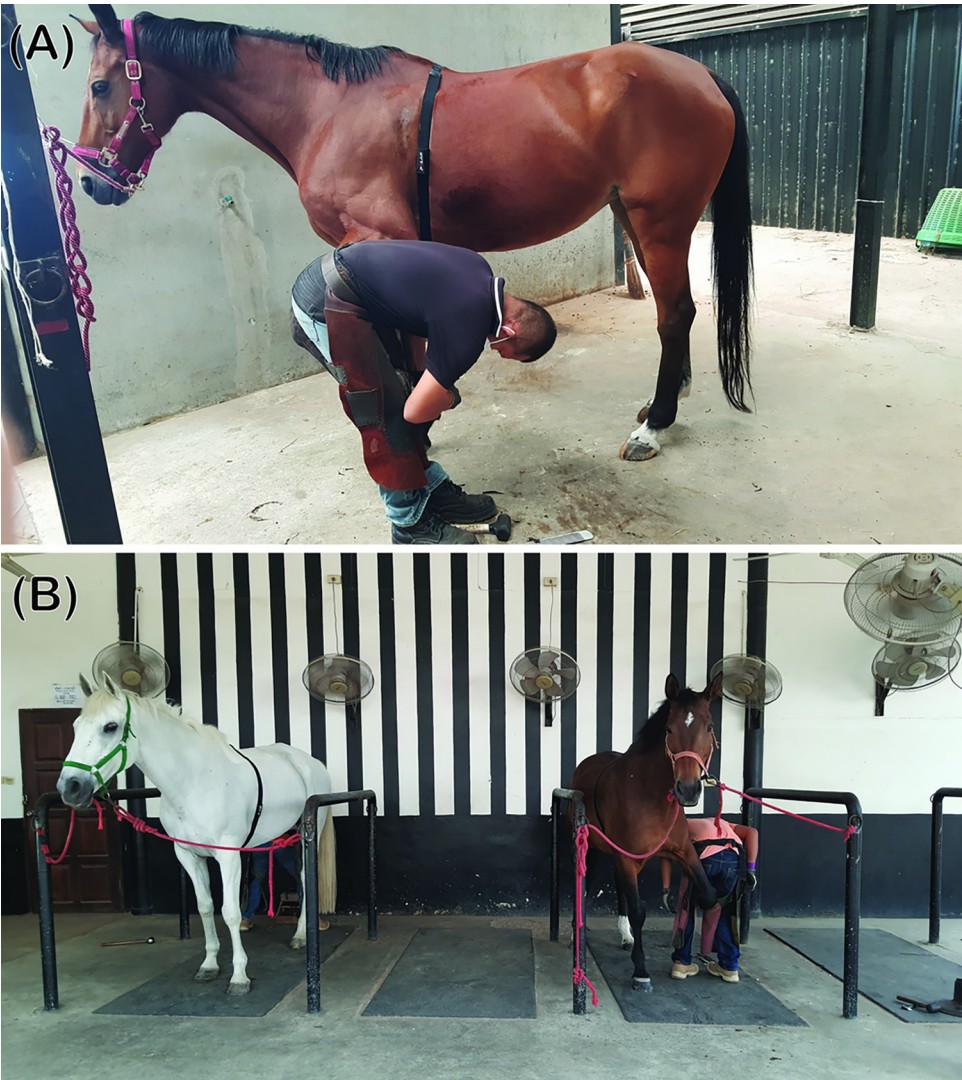

**Fig 2.** (A, B) Horses undergo the cold shoeing protocol while HRM devices are fastened around their chests.

intervals, HR, SDNN, RMSSD, pNN50, TINN, HRV triangular index, and stress index), (2) frequency domain variables (very-low-frequency [VLF] band [by default 0–0.04 Hz], LF band [by default 0.04–0.15 Hz], HF band [by default 0.15–0.4 Hz], LF/HF ratio, total power, VLF/total power ratio, LF/total power ratio, HF/total power ratio, and respiratory rate [RESP]), (3) nonlinear variables (SD1 and SD2), and (4) ANS (SD1% = [SD1 / (SD1 + SD2)] × 100, SD2% = [SD2 / (SD1 + SD2)] × 100), PNS, and SNS indexes. All HRV variables were determined 30 minutes before shoeing, during shoeing (the start of trimming until completed shoeing), and at 30-minute intervals for 120 minutes after shoeing (i.e., 30 minutes [0–30 minutes post-shoeing], 60 minutes [30–60 minutes post-shoeing], 90 minutes [60–90 minutes post-shoeing], and 120 minutes [90–120 minutes post-shoeing]).

## Statistical analysis

Statistical analyses were performed using GraphPad Prism software (version 10.1.0; GraphPad Software, Inc., San Diego, CA, USA). The Greenhouse–Geisser correction was automatically

applied to estimate an epsilon (sphericity) and correct for a lack of sphericity before analyses. The effects of the group, time, and group-by-time on the HRV variables were evaluated using a two-way repeated measures analysis of variance. HRV variables were compared within and between groups at a given time point using Tukey's multiple comparisons test. Data are expressed as the mean ± standard deviation (SD). Statistical significance was set at $p < 0.05$.

## Results

### Time domain variables

The changes in time domain variables in response to each shoeing method are shown in Table 1. Only time affected the changes in mean RR ($p = 0.0017$), mean HR ($p = 0.0062$), pNN50 ($p = 0.0005$), and stress index ($p < 0.0001$). However, the group-by-time interaction affected the changes in the SDNN ($p = 0.0002$), while the group-by-time interaction ($p = 0.0122$) and time ($p = 0.0477$) affected changes in the RMSSD. Notably, group and time separately affected changes in the HRV triangular index (group: $p = 0.0099$, time: $p < 0.0001$) and TINN (group: $p = 0.0244$, time: $p < 0.0001$).

The horses' mean HR and RR did not change during the cold and hot shoeing protocols compared to their pre-shoeing values. The mean HR ($p < 0.01$–$0.0001$) was elevated and the mean RR ($p < 0.01$–$0.0001$) was shorter at 30–120 minutes post-shoeing, while pNN50 had increased at 120 minutes post-shoeing ($p < 0.05$). The stress index decreased during shoeing until 120 minutes post-shoeing ($p < 0.05$–$0.001$). The SDNN and RMSSD were increased 30 minutes after hot but not cold shoeing (both $p < 0.05$). The HRV triangular index and TINN increased during hot shoeing until 120 min post-shoeing (both $p < 0.05$–$0.0001$). In contrast, among horses undergoing cold shoeing, TINN only increased during shoeing ($p < 0.05$), and the HRV triangular index only increased at 120 minutes after shoeing ($p < 0.05$). The SDNN, HRV triangular index, and TINN were higher among horses undergoing hot shoeing than cold shoeing at 30–120 minutes post-shoeing (SDNN and HRV triangular index: $p < 0.01$ at 30 minutes and $p < 0.05$ at 60–120 minutes post-shoeing; TINN: $p < 0.05$ for all given time points).

### Frequency domain variables

The effects of the shoeing protocols on the frequency domain variables are shown in Table 2. While group, time, and the group-by-time interaction did not affect the HF band ($p = 0.1191$, $p = 0.1338$, and $p = 0.1024$, respectively), they did affect the LF band ($p = 0.0228$, $p = 0.0028$, and $p = 0.0010$, respectively) and total power alteration ($p = 0.0305$, $p = 0.0408$, and $p = 0.0021$, respectively). Only time and the group-by-time interaction affected the VLF band ($p = 0.0462$ and $p = 0.0203$, respectively), LF/HF ratio ($p = 0.0162$ and $p = 0.0225$, respectively), and RESP ($p = 0.0143$ and $p = 0.0046$, respectively). Group and time independently affected the VLF/ total power ratio ($p = 0.0279$ and $p < 0.0001$, respectively), while only time affected the LF/ total power ratio ($p < 0.0001$). Group, time, and the group-by-time interaction did not affect the HF/total power ratio ($p = 0.3772$, $p = 0.8812$, and $p = 0.4004$, respectively).

The VLF variable did not change in horses undergoing cold or hot shoeing throughout the studied period. In contrast, the LF variable increased at 30 and 60 minutes after hot but not cold shoeing (both $p < 0.05$). The total power spectrum increased at 30 minutes after hot but not cold shoeing ($p < 0.05$). The LF/HF ratio decreased at 30 minutes after cold shoeing ($p < 0.05$) but did not change throughout the studied period with hot shoeing. The VLF/total power ratio decreased during hot shoeing until 120 minutes post-shoeing ($p < 0.05$–$0.01$); in contrast, it decreased at 60 and 120 minutes after cold shoeing (both $p < 0.01$). The LF/total power ratio was elevated at 30–120 minutes post-shoeing with both shoeing protocols ($p < 0.05$–$0.0001$). RESP was higher at 30 minutes after hot shoeing than during shoeing ($p < 0.05$). In contrast,

**Table 1. Time domain results (mean ± SD) for horses undergoing hot and cold shoeing.**

| HRV variables | Shoeing method | Pre-shoeing | During shoeing | Post-shoeing | | | | Interaction p-value |
|---|---|---|---|---|---|---|---|---|
| | | | | 30 min | 60 min | 90 min | 120 min | |
| Mean RR (ms)‡ | HSH | 1616.04 ± 290.60[ab] | 1705.37 ± 215.98[a] | 1570.48 ± 211.75[b] | 1565.26 ± 212.17[b] | 1562.22 ± 244.74[b] | 1590.65 ± 268.83[b] | 0.9291 |
| | CSH | 1706.96 ± 204.06[ab] | 1744.00 ± 185.73[a] | 1641.04 ± 187.68[b] | 1638.08 ± 160.79[b] | 1611.21 ± 166.19[b] | 1621.83 ± 197.45[b] | |
| Mean HR (bpm)‡ | HSH | 38.30 ± 7.67[ab] | 35.74 ± 4.85[a] | 38.83 ± 5.25[b] | 39.17 ± 6.29[b] | 39.26 ± 5.88[b] | 38.65 ± 6.30[b] | 0.8694 |
| | CSH | 35.71 ± 4.65[ab] | 34.88 ± 3.96[a] | 37.08 ± 4.29[b] | 36.92 ± 3.60[b] | 37.63 ± 3.91[b] | 37.505 ± 4.78[b] | |
| SDNN (ms) | HSH | 69.77 ± 30.92[b] | 68.37 ± 17.46[ab] | 86.44 ± 35.26[a,x] | 83.42 ± 36.74[ab,x] | 83.61 ± 34.01[ab,x] | 85.47 ± 29.92[ab,x] | 0.0002* |
| | CSH | 74.56 ± 33.24[a] | 67.01 ± 14.64[a] | 62.60 ± 13.56[a,y] | 63.80 ± 16.72[a,y] | 65.58 ± 17.11[a,y] | 69.00 ± 21.95[a,y] | |
| RMSSD (ms)‡ | HSH | 74.65 ± 45.00[b] | 74.33 ± 23.09[ab] | 97.43 ± 58.55[a,x] | 94.92 ± 59.59[ab] | 94.18 ± 54.94[ab] | 92.09 ± 40.86[ab] | 0.0122* |
| | CSH | 70.37 ± 33.34[a] | 67.31 ± 23.44[a] | 64.94 ± 21.17[a,y] | 68.30 ± 25.18[a] | 69.59 ± 26.46[a] | 74.84 ± 36.91[a] | |
| pNN50 (%)‡ | HSH | 36.51 ± 18.03[a] | 39.82 ± 12.31[a] | 45.53 ± 14.04[ab] | 45.73 ± 17.27[ab] | 45.63 ± 13.88[ab] | 46.37 ± 13.86[b] | 0.0767 |
| | CSH | 34.27 ± 15.82[a] | 34.29 ± 14.36[a] | 33.60 ± 12.83[ab] | 36.52 ± 13.99[ab] | 37.58 ± 14.96[ab] | 39.21 ± 15.95[b] | |
| TINN (ms)†,‡ | HSH | 355.13 ± 152.69[a] | 470.96 ± 129.19[b] | 484.13 ± 145.84[b,x] | 487.91 ± 202.92[b,x] | 482.65 ± 183.19[b,x] | 495.17 ± 143.66[b,x] | 0.3663 |
| | CSH | 332.96 ± 129.38[b] | 426.33 ± 75.41[a] | 397.83 ± 73.67[ab,y] | 392.25 ± 81.92[ab,y] | 395.46 ± 93.64[ab,y] | 418.08 ± 127.47[ab,y] | |
| HRV triangular index†,‡ | HSH | 12.70 ± 4.31[a] | 15.42 ± 3.56[b] | 18.44 ± 7.02[bc,x] | 18.49 ± 7.15[bc,x] | 17.60 ± 5.39[bc,x] | 18.83 ± 5.35[c,x] | 0.0869 |
| | CSH | 11.71 ± 4.80[a] | 14.01 ± 3.63[ab] | 13.85 ± 3.27[ab,y] | 14.76 ± 3.87[ab,y] | 14.69 ± 3.70[ab,y] | 15.19 ± 4.23[b,y] | |
| | | | | | | | | 0.6210 |
| Stress index‡ | HSH | 5.40 ± 2.14[a] | 4.10 ± 1.04[b] | 4.03 ± 1.78[b] | 4.10 ± 1.19[b] | 4.10 ± 1.31[b] | 3.90 ± 1.04[b] | |
| | CSH | 5.46 ± 2.34[a] | 4.32 ± 0.87[b] | 4.63 ± 0.92[b] | 4.63 ± 0.94[b] | 4.75 ± 1.21[b] | 4.53 ± 1.25[b] | |

The different letters a, b, and c indicate the statistical significance of the variable within the group. The different letters x and y indicate the statistical significance of the variable between the groups at the given time points.

* indicates a statistically significant group-by-time interaction.

† and ‡ represent significant group and time effects, respectively. **HSH**: Hot shoeing, **CSH**: Cold shoeing, **HRV**: Heart rate variability, **RR**: Beat-to-beat interval, **HR**: Heart rate, **SDNN**: Standard deviation of normal-to-normal RR intervals, **RMSSD**: Root mean square of successive RR interval differences, **pNN50**: Relative number of successive RR interval pairs that differ by >50 ms, **TINN**: Triangular interpolation of normal-to-normal intervals.

RESP was increased at 90 ($p < 0.01$) and 120 ($p < 0.05$) minutes after cold shoeing. The VLF/total power ratio differed significantly between the hot and cold shoeing protocols at 30 minutes post-shoeing ($p < 0.05$). The LF band and total power spectrum were higher with hot shoeing than cold shoeing at 30–120 minutes post-shoeing (LF: $p < 0.01$ at 30 minutes and $p < 0.05$ at 60–120 minutes post-shoeing; total power: $p < 0.05$ for all given time points).

## Nonlinear variables

Changes in SD1 and SD2 were not independently affected by group ($p = 0.0670$ and $p = 0.0597$, respectively) or time ($p = 0.0552$ and $p = 0.2711$, respectively). However, the group-

**Table 2. Frequency domain results (mean ± SD) for horses undergoing hot and cold shoeing.**

| HRV variables | Shoeing method | Pre-shoeing | During shoeing | Post-shoeing | | | | Interaction p-value |
|---|---|---|---|---|---|---|---|---|
| | | | | 30 min | 60 min | 90 min | 120 min | |
| VLF (ms²)‡ | HSH | 1066.8 ± 772.08[a] | 975.48 ± 547.53[a] | 1060.22 ± 629.16[a] | 934.74 ± 544.36[a] | 1058.35 ± 556.66[a] | 1141.22 ± 737.08[a] | 0.0203* |
| | CSH | 1506.9 ± 1473.91[a] | 1081.67 ± 587.13[a] | 868.50 ± 390.98[a] | 777.00 ± 417.00[a] | 867.88 ± 527.23[a] | 852.58 ± 472.69[a] | |
| LF (ms²)†,‡ | HSH | 2369.7 ± 2482.1[b] | 2241.22 ± 1226.10[bc] | 4051.17 ± 2682.40[a,x] | 4051.04 ± 3832.67[ac,x] | 4123.87 ± 3870.15[ab,x] | 4644.57 ± 4411.29[ab,x] | 0.0010* |
| | CSH | 2433.2 ± 1940.77[a] | 1997.38 ± 1001.66[a] | 1973.92 ± 893.13[a,y] | 2197.96 ± 1204.51[a,y] | 2217.54 ± 1248.99[a,y] | 2500.67 ± 1677.01[a,y] | |
| HF (ms²) | HSH | 1135.7 ± 1981.80[a] | 942.70 ± 676.19[a] | 2356.39 ± 4510.47[a] | 2256.30 ± 4405.33[a] | 2133.13 ± 4080.12[a] | 1494.78 ± 1397.07[a] | 0.1024 |
| | CSH | 682.92 ± 649.56[a] | 756.33 ± 561.58[a] | 778.67 ± 580.03[a] | 860.63 ± 730.92[a] | 887.67 ± 746.05[a] | 1105.46 ± 1312.85[a] | |
| LF/HF ratio‡ | HSH | 3.36 ± 2.94[a] | 2.80 ± 1.48[a] | 2.97 ± 1.40[a] | 3.26 ± 1.91[a] | 3.16 ± 1.99[a] | 3.56 ± 1.87[a] | 0.0225* |
| | CSH | 5.23 ± 3.80[a] | 3.81 ± 2.41[ab] | 3.17 ± 1.32[b] | 3.31 ± 1.72[ab] | 3.35 ± 1.80[ab] | 3.68 ± 2.64[ab] | |
| Total power (ms²)†,‡ | HSH | 4572.7 ± 4296.95[a] | 4159.70 ± 2188.60[ab] | 7468.35 ± 6872.07[b,x] | 7242.39 ± 7295.36[ab,x] | 7316.09 ± 6819.03[ab,x] | 7282.17 ± 5907.93[ab,x] | 0.0021* |
| | CSH | 4623.2 ± 3463.81[a] | 3835.25 ± 1629.77[a] | 3621.17 ± 1528.43[a,y] | 3835.79 ± 1920.67[a,y] | 3973.17 ± 2071.43[a,y] | 4458.79 ± 2868.42[a,y] | |
| VLF/Total power ratio†,‡ | HSH | 0.27 ± 0.13[a] | 0.23 ± 0.08[b,x] | 0.17 ± 0.07[b,x] | 0.17 ± 0.08[b] | 0.19 ± 0.08[b] | 0.18 ± 0.06[b] | 0.3493 |
| | CSH | 0.30 ± 0.14[a] | 0.29 ± 0.11[a,y] | 0.26 ± 0.09[a,y] | 0.22 ± 0.10[b] | 0.23 ± 0.11[ab] | 0.22 ± 0.09[b] | |
| LF/Total power ratio‡ | HSH | 0.49 ± 0.11[a] | 0.53 ± 0.07[a] | 0.58 ± 0.10[b] | 0.58 ± 0.12[b] | 0.57 ± 0.13[b] | 0.61 ± 0.09[b] | 0.2256 |
| | CSH | 0.51 ± 0.14[a] | 0.51 ± 0.07[a] | 0.54 ± 0.07[b] | 0.56 ± 0.08[b] | 0.56 ± 0.07[b] | 0.56 ± 0.09[b] | |
| HF/Total power ratio | HSH | 0.24 ± 0.16[a] | 0.24 ± 0.11[a] | 0.24 ± 0.12[a] | 0.25 ± 0.14[a] | 0.24 ± 0.13[a] | 0.21 ± 0.10[a] | 0.4004 |
| | CSH | 0.20 ± 0.19[a] | 0.20 ± 0.13[a] | 0.20 ± 0.08[a] | 0.22 ± 0.11[a] | 0.21 ± 0.10[a] | 0.23 ± 0.15[a] | |
| RESP (Hz)‡ | HSH | 0.220 ± 0.047[ab] | 0.196 ± 0.042[a] | 0.220 ± 0.022[b,x] | 0.223 ± 0.037[ab] | 0.214 ± 0.047[ab] | 0.207 ± 0.035[ab] | 0.0046* |
| | CSH | 0.194 ± 0.034[a] | 0.201 ± 0.028[a] | 0.205 ± 0.021[ac,y] | 0.216 ± 0.019[abc] | 0.223 ± 0.018[b] | 0.222 ± 0.023[c] | |

The different letters a, b, and c indicate the statistical significance of the variables within the group. The different letters x and y indicate the statistical significance of the variables between the groups at the given time points.

* indicates a statistically significant group-by-time interaction.

† and ‡ indicate significant group and time effects, respectively. **HSH**: Hot shoeing, **CSH**: Cold shoeing, **VLF**: HRV very-low-frequency band (by default 0–0.04 Hz), **LF**: HRV low-frequency band (by default 0.04–0.15 Hz), **HF**: HRV high-frequency band (by default 0.15–0.4 Hz), **RESP**: Respiratory rate.

by-time interaction affected SD1 ($p = 0.0112$) and SD2 ($p = 0.0001$). Only time affected the SD2/SD1 ratio ($p = 0.0495$). Regarding the shoeing protocols, SD1 and SD2 did not change with cold shoeing throughout the studied period. While SD1 and SD2 did not change during the hot shoeing protocol, they were elevated at 30 minutes post-shoeing (both $p < 0.05$). SD1 differed significantly between the hot and cold shoeing protocols at 30 minutes post-shoeing ($p < 0.05$), while SD1 differed nonsignificantly at 60 ($p = 0.0570$) and 90 ($p = 0.0608$) minutes after hot shoeing. In contrast, SD2 differed significantly between hot and cold shoeing methods at 30–120 minutes post-shoeing ($p < 0.01$ at 30 minutes and $p < 0.05$ at 60–120 minutes

**Table 3. Nonlinear results and ANS indexes (mean ± SD) for horses undergoing hot and cold shoeing.**

| HRV variables | Shoeing method | Pre-shoeing | During shoeing | Post-shoeing | | | | Interaction p-value |
|---|---|---|---|---|---|---|---|---|
| | | | | 30 min | 60 min | 90 min | 120 min | |
| **Nonlinear results** | | | | | | | | |
| SD1 (ms) | HSH | 52.96 ± 32.00[a] | 52.57 ± 16.35[ab] | 68.92 ± 41.42[b,x] | 67.17 ± 42.16[ab] | 66.67 ± 38.90[ab] | 65.16 ± 28.91[ab] | 0.0112* |
| | CSH | 50.32 ± 24.20[a] | 47.62 ± 16.58[a] | 45.98 ± 14.98[a,x] | 48.32 ± 17.82[a] | 49.23 ± 18.73[a] | 52.95 ± 26.11[a] | |
| SD2 (ms) | HSH | 82.04 ± 33.42[ab] | 80.50 ± 21.32[a] | 99.59 ± 32.60[b,x] | 95.50 ± 34.97[ab,x] | 96.37 ± 32.53[ab,y] | 100.74 ± 34.41[ab,x] | 0.0001* |
| | CSH | 90.85 ± 43.18[a] | 80.50 ± 19.34[a] | 75.24 ± 14.65[a,y] | 75.44 ± 18.90[a,y] | 77.78 ± 19.31[a,y] | 80.57 ± 22.79[a,y] | |
| SD2/SD1[‡] | HSH | 1.72 ± 0.56[a] | 1.59 ± 0.38[a] | 1.58 ± 0.37[a] | 1.59 ± 0.45[a] | 1.57 ± 0.34[a] | 1.64 ± 0.39[a] | 0.4848 |
| | CSH | 1.91 ± 0.69[a] | 1.85 ± 0.60[a] | 1.73 ± 0.36[a] | 1.65 ± 0.39[a] | 1.70 ± 0.48[a] | 1.68 ± 0.49[a] | |
| **ANS indexes** | | | | | | | | |
| SD1% | HSH | 38.31 ± 7.89[a] | 39.47 ± 5.58[a] | 39.54 ± 6.06[a] | 39.76 ± 6.78[a] | 39.64 ± 5.84[a] | 38.75 ± 6.28[a] | 0.7364 |
| | CSH | 36.57 ± 9.71[a] | 36.74 ± 8.62[a] | 37.32 ± 4.97[a] | 38.59 ± 6.28[a] | 38.12 ± 6.49[a] | 38.63 ± 7.80[a] | |
| SD2% | HSH | 61.69 ± 7.89[a] | 60.54 ± 5.58[a] | 60.46 ± 6.06[a] | 60.24 ± 6.78[a] | 60.36 ± 5.84[a] | 61.24 ± 6.28[a] | 0.7364 |
| | CSH | 63.43 ± 9.71[a] | 63.26 ± 8.62[a] | 62.68 ± 4.97[a] | 61.41 ± 6.28[a] | 61.88 ± 6.49[a] | 61.38 ± 7.80[a] | |
| PNS index | HSH | 3.98 ± 2.08[a] | 4.45 ± 1.33[a] | 4.44 ± 1.98[a] | 4.37 ± 1.95[a] | 4.33 ± 2.06[a] | 4.38 ± 2.02[a] | 0.2556 |
| | CSH | 4.29 ± 2.00[a] | 4.37 ± 1.21[a] | 3.86 ± 1.10[a] | 3.95 ± 1.12[a] | 3.86 ± 1.21[a] | 4.05 ± 1.57[a] | |
| SNS index[‡] | HSH | −2.40 ± 0.73[b] | −2.78 ± 0.42[a] | −2.60 ± 0.47[b] | −2.57 ± 0.53[b] | −2.56 ± 0.52[b] | −2.62 ± 0.56[ab] | 0.7896 |
| | CSH | −2.54 ± 0.44[b] | −2.78 ± 0.35[a] | −2.59 ± 0.35[b] | −2.61 ± 0.36[b] | −2.55 ± 0.41[b] | −2.59 ± 0.48[ab] | |

The different letters a, b, and c indicate the statistical significance of the variables within the group. The different letters x and y indicate the statistical significance of the variables between the groups at the given time points.

* indicates a statistically significant group-by-time interaction.

[‡] denotes a significant time effect. **HSH**: Hot shoeing, **CSH**: Cold shoeing, **SD1**: Standard deviation of the Poincaré plot perpendicular to the line of identity, **SD2**: Standard deviation of the Poincaré plot along the line of identity, **PNS**: Parasympathetic nervous system, **SNS**: Sympathetic nervous system.

post-shoeing). However, the SD2/SD1 ratio did not change throughout the studied period with the hot or cold shoeing protocols (Table 3).

## ANS indexes

Group, time, and the group-by-time interaction did not affect the SD1% and SD2% ($p = 0.3610$, $p = 0.3709$, and $p = 0.7364$, respectively, for both variables) or the PNS index ($p = 0.5098$, $p = 0.5802$, and $p = 0.2556$, respectively). While group ($p = 0.8549$) and the group-by-time interaction ($p = 0.7896$) did not significantly affect the SNS index, time did significantly affect the SNS index ($p = 0.0051$). The SNS index only decreased during shoeing ($p < 0.001$) and returned to a value similar to pre-shoeing at 30–120 minutes post-shoeing (Table 3).

## Discussion

This study investigated and compared the effects of different shoeing methods on HRV in horses. It found (1) the interaction between the shoeing method and time affected multiple HRV variables, including SDNN, RMSSD, the VLF band, the LF band, the LF/HF ratio, RESP, total power, SD1, and SD2; (2) the shoeing method and time independently affected TINN, HRV triangular index, and the VLF/total power ratio; (3) SDNN, RMSSD, the LF band, total power, SD1, and SD2 were increased at 30 minutes after hot shoeing; (4) HRV triangular index and VLF/total power ratio were increased during hot shoeing until 120 minutes after hot shoeing; (5) SDNN, TINN, HRV triangular index, LF, total power, and SD2 were higher in horses undergoing hot shoeing than cold shoeing at 30–120 minutes post-shoeing. These results suggest that each shoeing method exerts a particular effect on HRV. An increase in HRV was detected mainly for 30 minutes in horses after hot shoeing. Higher HRV variables reflected greater sympathovagal activity in horses undergoing hot shoeing than cold shoeing at 30–120 minutes post-shoeing.

HRV has been used to determine SNS and PNS oscillations during the cardiac cycle [11,21,32]. An increase in sympathetic activity results in an elevated HR and reduced HRV, whereas an increase in vagal activity results in decreased HR and increased HRV [11,32]. The HRV variables computed before shoeing did not differ significantly between hot and cold shoeing groups (Tables 1–3), indicating that horses in both groups showed similar autonomic responses before the study commenced. According to autonomic regulation, the HR at a given point in time generally reflects the net sympathetic and vagal regulation activities [21]. An increase in HR is associated with increased sympathetic activity [33], while a decrease in HR is related to increased vagal activity [21]. While HR is a commonly used parameter to indicate stress [19,34] and exercise performance [35], it might be an unreliable indicator of ANS activity [21] since the two branches of the ANS may not act as a continuum when an increase in one component coincides with a decrease in another during cardiac regulation [21,32]. Instead, they can act on cardiac function independently or synchronously with each autonomic nerve branch, leading to multiple potential nerve activation patterns during cardiac cycle regulation. In this study, the increase in HR after shoeing compared to during shoeing may have resulted from a synchronously increased sympathetic impulse corresponding to decreased vagal activity or asynchronous action in sympathetic and vagal components. Therefore, using a simple HR measurement may be inappropriate to determine the characteristics of ANS impulses during the functional activity of the heart [32].

Regarding hoof conformation, the hoof wall grows over time. The increased weight-bearing length following toe growth and a decreased heel angle may cause hoof angle displacement [25]. Moreover, since metal shoes do not change in size, the maximum expansion of the hoof wall is limited. These events cause gradual discomfort in the shod feet of horses. Routine hoof trimming and shoeing are expected to offer horses comfort after shoeing. This study used several time domain measures to determine ANS activity after shoeing. The pNN50 showed an increasing trend at 30–90 minutes post-shoeing and a significant increase at 120 minutes post-shoeing with both hot and cold shoeing. Since an increase in pNN50 has been reported to indicate predominant vagal tone [21], the gradual increase in pNN50 may indicate an increase in vagal tone in horses from before to after shoeing. Furthermore, the stress index was lower during shoeing until 120 minutes post-shoeing with both hot and cold shoeing. These results suggested that horses generally experienced more comfort after both hot and cold shoeing.

While the HRV time domains demonstrated the effects of sympathetic and vagal activities on cardiac cycle regulation, a lack of interplay between SNS and PNS impulses makes them insufficient to reflect the sympathovagal balance and overall ANS activity [21,36]. Therefore,

sympathovagal balance based solely on time domain parameters must be interpreted cautiously [9]. The frequency domain results can indicate integral sympathetic and vagal activities and, in turn, sympathovagal balance in horses [18,21]. The frequency band thresholds used in this study were adopted from those in humans, which have also been applied in equine research [10,24]. The HF band is majorly affected by vagal nerve impulses and the RESP of the horse. Therefore, HF power indicates cardiac vagal activity [18]. In contrast, LF power changes with the influence of both sympathetic and vagal activities on blood pressure regulation [18,21]. While HF power showed an increasing trend at 30–90 minutes after hot shoeing (S1 Fig), the considerable deviation in its values caused a nonsignificant change. The LF band and total power increased markedly at 30 minutes post-hot shoeing, and the LF band was also higher in hot-shoed horses than in cold-shoed horses throughout the 120 minutes post-shoeing. These results indicate a larger contribution of sympathovagal activity in hot-shoed horses than in cold-shoed horses after shoeing.

Moreover, an elevated LF/total power ratio, corresponding to a decreased VLF/total power ratio and unchanged HF/total power ratio, may at least partially reflect a significant contribution of the LF band to total power during the 120 minutes post-shoeing. The contributions of sympathetic and vagal activities to SDNN are highly correlated with the VLF band, the LF band, and total power [37]. This study's results are consistent with the abovementioned study, revealing a corresponding increase in SDNN, the LF band, and total power and supporting the contribution of sympathetic and vagal activities during the entire post-hot shoeing period (Table 2).

Since the HF band is influenced by the respiratory cycle, an increase in RESP leads to an increase in the HF band [9,11], especially during intense exercise [18]. However, in this study, RESP did not correspond to changes in the HF band but corresponded, in part, to the change in the LF band 30 minutes after hot shoeing. Since the RR intervals were recorded when the shod horse was at rest, it is plausible that vagal nerve impulses quickly generated the rhythmic oscillation of the heart crossing over into the LF band, similar to the descriptions in previous studies [11,38,39]. Therefore, respiration-related vagal activity may, at least in part, influence changes in the LF band in horses 30 minutes after hot shoeing.

The LF/HF ratio of the power spectrum also appears to be a helpful indicator of sympathetic impulses during physical and psychological stress [40]. Specifically, an increase in the LF/HF ratio indicates predominant sympathetic activity, especially in fight/flight/freeze behavior [11]. Notably, an unchanged LF/HF ratio throughout the hot shoeing period can be considered evidence of the integral contribution of sympathetic and vagal impulses in response to this protocol in these horses.

Nonlinear variables were also determined to evaluate ANS generation in response to the shoeing protocols. In this study, an increase in SD1 was consistent with an increase in RMSSD, indicating an increase in vagal activity in horses 30 minutes after hot shoeing. Furthermore, SD2 increased similarly to SDNN and the LF band 30–120 minutes after hot shoeing. These results support more pronounced vagal activity at 30 minutes, followed by increased sympathovagal activity 60–120 minutes after hot shoeing. In addition, the lack of change in the SD2/SD1 ratio, which corresponds to the unchanged LF/HF ratio, throughout the studied period indicates the equal contributions of sympathetic and vagal activities at given times in horses undergoing hot shoeing.

The interaction between the shoeing protocol and time affected several HRV variables: SDNN, RMSSD, the VLF band, the LF band, the LF/HF ratio, RESP, total power, SD1, and SD2. These findings indicate that the two shoeing methods have distinct effects on ANS responses. Since the HRV variables did not differ significantly between hot and cold shoeing before and during shoeing, it was concluded that different shoeing methods provoked similar

ANS responses during shoeing. Nevertheless, the changes in HRV variables were mainly seen at 30 minutes post-shoeing, particularly in hot-shoed horses. Moreover, SDNN, TINN, HRV triangular index, LF band, total power, and SD2 were higher in hot-shoed horses than cold-shoed horses throughout the 120 minutes post-shoeing.

The findings of this study seem to confirm that hot shoeing is more beneficial for horses than cold shoeing [1]. The study found that hot shoeing improves sympathovagal activity and HRV, providing objective evidence for its benefits. Since shoeing affects a horse's gait, it can also affect the load on the lower limb, which can have a negative impact on the horse's locomotion [41]. Long-term front limb shoeing can cause morphological hoof changes that may lead to lameness [5,42]. Therefore, it is necessary to continue studying the effects of shoeing on HRV variables during ordinary locomotion and exercise in athletic horses.

It should be noted that the present study had certain limitations. Firstly, a cross-over design was not used because shoeing horses with an unfamiliar method might cause stress, which could lead to non-representative changes in HRV after shoeing. Additionally, due to the individual characteristics of each horse and the farrier's particular style, the duration of individual shoeing could not be strictly controlled. The baseline measurements of HRV were taken from the designated stables at the farrier place, which may also be considered as a stressor before shoeing. Therefore, the modification of HRV before and during shoeing should be interpreted with caution. Furthermore, no behavioral evaluation was carried out during the study, which resulted in a lack of additional evidence to determine stress during shoeing protocols. Finally, it is important to note that significant inter-individual variations were observed due to differences in housing location, horse breed, sex, and equestrian sport discipline. This could be the underlying cause of significant deviations in the HRV variables among horses for both shoeing protocols.

## Conclusions

ANS activity responds differently in horses undergoing different shoeing methods, with greater HRV 30–120 minutes after hot shoeing compared to cold shoeing. More research is needed to substantiate whether these differences can be attributed to the fact that horses experience more comfort from hot shoeing compared to cold shoeing.

## Supporting information

**S1 Fig. The high-frequency domain of heart rate variability in horses undergoing hot and cold shoeing.** A trend toward an increase in the high-frequency domain is observed at 30–90 min post-shoeing in horses undergoing hot shoeing.
(DOCX)

**S1 Table. The horses with different disciplines that underwent hot or cold shoeing.**
(DOCX)

## Acknowledgments

We thank Harald Link, the president of the Thailand Equestrian Federation (TEF); Nunthinee Tanner, the vice president of the TEF; and the Thai Polo and Equestrian Club for allowing us to measure the HRV variables of horses undergoing cold shoeing. We also thank Vaewratt Kamonkon, director of the Horse Lover's Club; Chalermchan Yosviriyapanich, director of the House of Horse Riding Club; and Ploypailin Pattanakul for allowing us to measure the HRV variables of horses undergoing hot shoeing. Finally, we thank Khunanont Thongcham and Chanikarn Srirattanamongkol for collecting data during the cold shoeing protocol.

## Author Contributions

**Conceptualization:** Onjira Huangsaksri, Thita Wonghanchao, Kanokpan Sanigavatee, Chanoknun Poochipakorn, Metha Chanda.

**Data curation:** Onjira Huangsaksri, Thita Wonghanchao, Kanokpan Sanigavatee, Chanoknun Poochipakorn, Metha Chanda.

**Formal analysis:** Onjira Huangsaksri, Thita Wonghanchao, Kanokpan Sanigavatee, Chanoknun Poochipakorn, Metha Chanda.

**Funding acquisition:** Onjira Huangsaksri, Thita Wonghanchao, Kanokpan Sanigavatee, Metha Chanda.

**Investigation:** Onjira Huangsaksri, Thita Wonghanchao, Kanokpan Sanigavatee, Chanoknun Poochipakorn, Metha Chanda.

**Methodology:** Onjira Huangsaksri, Thita Wonghanchao, Kanokpan Sanigavatee, Chanoknun Poochipakorn, Metha Chanda.

**Project administration:** Chanoknun Poochipakorn, Metha Chanda.

**Resources:** Thita Wonghanchao, Chanoknun Poochipakorn, Metha Chanda.

**Software:** Kanokpan Sanigavatee, Metha Chanda.

**Supervision:** Chanoknun Poochipakorn, Metha Chanda.

**Validation:** Onjira Huangsaksri, Thita Wonghanchao, Kanokpan Sanigavatee, Chanoknun Poochipakorn, Metha Chanda.

**Visualization:** Onjira Huangsaksri, Thita Wonghanchao, Kanokpan Sanigavatee, Chanoknun Poochipakorn, Metha Chanda.

**Writing – original draft:** Onjira Huangsaksri, Thita Wonghanchao, Metha Chanda.

**Writing – review & editing:** Metha Chanda.

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
