## [Decision Letter · Decision Letter 0]

2 Oct 2023

PONE-D-23-24799Heart rate and heart rate variability in horses receiving hot shoeing and cold shoeing protocolsPLOS ONE

Dear Dr. Chanda,

Thank you for submitting your manuscript to PLOS ONE. After careful consideration, we feel that it has merit but does not fully meet PLOS ONE’s publication criteria as it currently stands. Therefore, we invite you to submit a revised version of the manuscript that addresses the points raised during the review process.

The presented manuscript relates to a study on the effect of two shoeing methods on HRV parameters in horses. The aims of the study are of interest however it is not clear whether the study intends to investigate psychological stress and/or physiological stress. The subject of the submitted draft is within the scope of PLOS ONE. The manuscript is well written. The introduction is well composed and introduces well the question. The Materials and Methods section could be more detailled to present the conditions in which the measurements were performed. Although the used statistical analysis seems appropriate, more explanation on whether the requirements for performing this two-way ANOVA were met. The Results section could be more synthesized in the text. The Discussion section is well written and the author did discuss well the results of the study, including the limitations.The authors should comment on all the issues risen by the reviewers.

We look forward to receiving your revised manuscript.

Kind regards,

Nejka Potocnik

Academic Editor

PLOS ONE

 “This work is supported, in part, by financial support From Kasetsart Veterinary Development Funds.”

3. We note that you have stated that you will provide repository information for your data at acceptance. Should your manuscript be accepted for publication, we will hold it until you provide the relevant accession numbers or DOIs necessary to access your data. If you wish to make changes to your Data Availability statement, please describe these changes in your cover letter and we will update your Data Availability statement to reflect the information you provide

5. We note that Figures 1 and 2 in your submission contain copyrighted images. All PLOS content is published under the Creative Commons Attribution License (CC BY 4.0), which means that the manuscript, images, and Supporting Information files will be freely available online, and any third party is permitted to access, download, copy, distribute, and use these materials in any way, even commercially, with proper attribution. For more information, see our copyright guidelines: http://journals.plos.org/plosone/s/licenses-and-copyright.

a. You may seek permission from the original copyright holder of Figures 1 and 2 to publish the content specifically under the CC BY 4.0 license.

Reviewers' comments:

Reviewer's Responses to Questions

**Comments to the Author**

1. Is the manuscript technically sound, and do the data support the conclusions?

Reviewer #1: Partly

Reviewer #2: Partly

2. Has the statistical analysis been performed appropriately and rigorously? 

Reviewer #1: Yes

Reviewer #2: No

3. Have the authors made all data underlying the findings in their manuscript fully available?

Reviewer #1: Yes

Reviewer #2: Yes

4. Is the manuscript presented in an intelligible fashion and written in standard English?

Reviewer #1: Yes

Reviewer #2: Yes

5. Review Comments to the Author

Reviewer #1: The presented manuscript relates to a study on the effect of 2 shoeing methods on equine stress. The aims of the study are well defined, of interest in the current era of awareness over animal welfare, and are well followed through the manuscript. The subject of the submitted draft is within the scope of PLOS ONE. The manuscript is well written. The introduction is well composed and introduces well the question. The Materials and Methods section could be more detailled to ensure the readers can understand what the investigators have performed (see below). The Results section could in my opinion be more synthesized in the text (as a lot of results are presented in the tables). The Discussion section is well written and the author did discuss well the results of the study, including the limitations.

Please find hereunder suggestions from my side and I would be happy to read and review a revised version of the manuscript.

Major

- As author mentionned, a crossover design should have been preferred to the current design, as different horses could have respond differently to treatment, independently from the treatment method. Such a design is possible, however with adequate statistical analysis showing the robustness of the set and the significance of the results.

- A lot of HRV-values were analyzed. To my knowledge, the use of some of them have not been validated in horses. Please specify on which time period the recordings were analysed. For time periods shorter than 5 minutes, some values could not be suitabe. Please discuss the repeatability of measurements, which could be of interest as bias for the observed differences observed in the post-shoeing. Please see van Vollenhoven et al. 2016 (https://doi.org/10.1016/j.jevs.2016.07.006) for HRV in rest and Frippiat et al. 2023 (https://doi.org/10.1163/17552559-20220044) for HRV during activities. The observed differences in 30, 60, 90, 120 min post shoeing should be discussed with the findings in those 2 studies.

- As authors may select only some HRV-variables, Results section could be more synthesized to keep the essential information which is not shown in the tables. Please review the Discussion section as many used HRV-variables relate to the same information about the autonomic nervous system. Several of theses variables have been described in horses in experimental conditions, while the latest publications over the observation of stress-responses rely on only a few of them. For the readiness and understanding of the reader, I advice reduce the amount of redundant information.

Minor suggestions

- Through the whole manuscript: time-domain and frequency-domain with a '-'

- Through the whole manuscript: use cold-shoeing and hot-shoeing

- Line 17: replace 'aims' to 'aimed'

- Line 18: delete 'shoeing' after 'cold' to avoid repetition

- Line 20: prefer 'before shoeing' to 'at pre-shoeing'

Abstract

- Lines 19-21: Please rephrase to show more clearly the study design: did all horses receiving both treatments in a crossover design? Specify the techniques for data acquisition.

- Lines 22-31: All these abbreviations have not been introduced, and do not sufficiently explain what the authors did show. Please rephrase these sentences to clarify what did change and what not, in both conditions.

- State once the level of the significance to avoid repetition of (p<0.05).

- Line 31: I presume authors mean parasympathetic activity (in this case, change the keyword at line 34)

Introduction

- Line 38-39: What do authors mean with short-term fluctuation of RR interval? Some HRV-parameters are related to short-term variability, while others do relate to the whole recording. Please specify more clearly.

- Line 41-42: Please specify how the variation aids in maintaining cardiovascular homeostasis and automatic responses to challenges.

- Line 46-66: Several sentences do belong to the Discussion Section and not the Introduction Section. Please do condensate the information to the needed one.

Materials and Methods

- Line 85: please specify the body weight of the horses

- Line 144-153: depending on the changes in the Introduction Section, abbreviations can be used without full text as they are introduced elsewhere

- Line 151: please introduce a reference for the validation of the use of respiratory rate in horses

- Line 161: please state how the normal distribution was assessed as author used mean and standard deviation

- Please add a study design. Did all horses follow both treatments on separate days?

Results

Line 192-193: HR and RR are 2 same data. Increase in HR implicates decrease in RR. Please choice one of the two.

Reviewer #2: The article presents an interesting and original study on the effects of shoeing methods that may benefit horse welfare. Although the used statistical analysis seems appropriate, more explanation on whether the requirements were met to perform this two-way ANOVA were met. Furthermore, it is unclear whether the study intends to investigate psychological stress and/or physiological stress and more information on the conditions in which the measurements were performed is needed for the several time periods.

6. PLOS authors have the option to publish the peer review history of their article (what does this mean?). If published, this will include your full peer review and any attached files.

Reviewer #1: No

Reviewer #2: No

---

## [Author Response · Author response to Decision Letter 0]

29 Nov 2023

The manuscript entitled “Heart rate and heart rate variability in horses receiving hot shoeing and cold shoeing protocols”

Dear Editor,

I’d appreciate your dedicated time in reviewing our manuscript and giving valuable comments to improve the context of this manuscript. We’ve addressed all points raised by the reviewers, and the track changes within the text noticed this modification.

Editor’s comments

The presented manuscript relates to a study on the effect of two shoeing methods on HRV parameters in horses. The aims of the study are of interest however it is not clear whether the study intends to investigate psychological stress and/or physiological stress. The subject of the submitted draft is within the scope of PLOS ONE. The manuscript is well written. The introduction is well composed and introduces well the question. The Materials and Methods section could be more detailled to present the conditions in which the measurements were performed. Although the used statistical analysis seems appropriate, more explanation on whether the requirements for performing this two-way ANOVA were met. The Results section could be more synthesized in the text. The Discussion section is well written and the author did discuss well the results of the study, including the limitations.

Response to the editor:

As per the editor’s question about the objective of this study, this study aims to investigate the physiological stress (autonomic response) in response to the usual shoeing protocols in horses. Generally, conventional shoeing methods by professional farriers usually offer horses comfort after shoeing. The difference in autonomic response in horses receiving shoeing should be more or less comfortable rather than higher or lower stress following shoeing protocol. After careful revision, we decided to use the term “comfort” instead of “stress” to reflect the outcome of shoeing protocols as it delivers a more explicit message from this study. We’ve modified the main text over this issue on page 1, line30-32, and page 20, lines 417 and 421. 

Additionally, we haven’t used the track change method in this revision to demonstrate where the text has been corrected. As there are a lot of corrections and modifications within the main text, we decided to get the full revised manuscript grammatically corrected by the Cambridge proofreading service (the proofreading certificate was attached to the resubmission system). So, the corrections and modifications according to the reviewers’ comments were highlighted in green.

Reviewers’ comments

Generally this article is a nice study on the effects of shoeing on stress in horses which may be relevant for the welfare of horses. However, it is unclear what definition is used of stress and what exactly is investigated. Therefore, the study does need some work and may benefit from clarification. Hopefully the comments below will aid in this process.

Response to the reviewer:

We’d like to add more information regarding Heart rate variability (HRV) modulation and stress response to stimulus. HRV has frequently been used for indicating stress response in horses during exercise (https://doi.org/10.3389/fphys.2016.00155), with pain (https://doi.org/10.1016/j.jveb.2017.09.002), and other stressors such as transport (https://doi.org/10.1016/j.yhbeh.2009.11.003).. An increase in HRV, indicated by increased several HRV variables, reflects parasympathetic dominance, more excellent adaptive capability, and lower stress in response to challenges. Regarding this study, we found that several HRV variables increased significantly during the hot shoeing protocol, lasting until 120 minutes post-shoeing. Moreover, horses receiving hot shoeing showed higher HRV than those with cold shoeing throughout the 120-minute post-hot shoeing period. This increased HRV reflects more body adaptability to cope with challenges. Therefore, those horses with hot shoeing are more comfortable than those with cold shoeing periods after shoeing. The sentences to deliver this message were modified on page 2, lines 30-32, and page 20, lines 417-421. 

More importantly, we have decided to use “more comfort” instead of “lower stress” following increased or higher HRV in horses receiving hot shoeing to avoid confusion and misconduct in the messages. That’s because the appropriate shoeing method typically causes the horse to be comfortable after shoeing. The increased HRV or higher HRV should mean more comfort rather than lower stress.

31: during which time period more sympathovagal activity?

Response to the reviewer:

We do apologize for giving the unsuitable word and misleading the reader with this term. In this sentence, we’ve changed to use the parasympathetic activity instead of the sympathovagal activity. This study found that the SDNN, TINN, HRV triangular index, LF power spectrum and SD2 increased during 30-120 minutes. So, the more sympathovagal activity in this context was found throughout 120 minutes post-shoeing in horses with hot shoes. We

32/33: Increases or decreases in sympathovagal activity do not necessarily mean differences between horses that are hot-shoed or cold-shoed. Can you draw the conclusion from you results that hot-shoeing provides more stress than cold-shoeing? Perhaps the results are better reflected by the sentence: hot-shoed horses experience stress whereas cold-shoed horses do not show the same response?

Response to the reviewer:

As explained above, appropriate conventional shoeing methods by professional farriers offer horses comfort, indicated by increased HRV in horses after shoeing. Higher HRV in horses after hot shoeing should be declared as more comfort rather than lower stress. So, “comfort” was used instead of “stress” in this context. We’d like to explain a little bit regarding the conclusion of this study. In the present study, an increase in mean RR, pNN50, TINN, HRV, and triangular index, for example, was observed in horses receiving both shoeing protocols, supposing that horses derive comfort after shoeing compared to before shoeing. However, the higher HRV after shoeing in horses with hot shoeing compared to cold shoeing. This result reflects an excellent sympathovagal action leading to higher HRV and mirroring more comfort in hot-shoed horses than cold-shoed horses after shoeing. This study concluded that horses derive more comfort after applying hot shoeing than cold shoeing.

No differences between pre and during shoeing?

Response to the reviewer:

There was no statistical difference between pre- and during shoeing in horses receiving both types of shoeing protocol. The difference was majorly noticed after shoeing. 

38: Does HRV only reflects short term fluctuations, or does that depend on the type of parameter you use?

Response to the reviewer:

We do apologize for giving the reviewers an unclear message over this issue. The HRV variables can reflect both short-term (RMSSD, NN50, pNN50, HF power spectrum and SD1 nonlinear results) and long-term fluctuation (mean RR, SDNN, TINN, HRV triangular index, LF power spectrum, SD2 nonlinear results) of heartbeats. To avoid misconducting this message, we’ve modified the sentence of the introduction part to “Heart rate variability (HRV) is the time difference between beat-to-beat (RR) intervals during the cardiac cycle” on page 3, lines 53-54.

39: does the fluctuation accompany oscillatory influences or does it reflect them?

Response to the reviewer:

We’d like to explain a little bit regarding the HRV modulation. The contraction and relaxation of the heart muscle in the cardiac cycle accompany the rhythmic impulse of the autonomic nervous system (ANS), including sympathetic and parasympathetic components, acting on the heart’s sinoatrial node. So, the variation of the time interval between heartbeats is under the oscillatory influences of both ANS components as well as reflects the synchronous activities of both parts.

43: what does the author use as indication of stress, is this physiological stress or mental stress? The literature referred to is mostly directed at physiological stress. If the author also would like to include psychological stress please include more literature on this.

Response to the reviewer:

We do apologize again for giving the reviewers an unclear definition of stress. It was physiological stress, as per the reviewer’s mention. We’ve modified it accordingly on page 3, lines 59-60.

64: is referring to human literature here relevant? If there is quite a substantial amount of literature on horses why would you include human literature?

Response to the reviewer:

Thank you for your question. In this connection, we agree with the reviewer there is a lot of literature regarding HRV in horses. However, we just want to address the use of the HRV parameter in humans in a few sentences because we’d like to convince the readers of the worldwide reliability of using the HRV parameter to indicate stress response in horses and humans. However, after careful consideration, we removed it from the text and left only the HRV modulation in horses in this manuscript. 

67: perhaps this Alinea (no. 3) should be moved to the start the article, as it become clear than that it covers the procedures of horseshoeing and why it is relevant to investigate this further 

Response to the reviewer:

Thank you for your suggestion. We’ve moved as well as modified the text of the shoeing protocol description to the first paragraph of the introduction part on page 3, lines 36-45.

71: “wildly”> I think the author means “widely”?

Response to the reviewer:

Thank you for your suggestion. We’ve revised it accordingly on page 3, line 39.

73: what does the author mean by “despite a positive outcome” that generally shoeing is better for the hoof condition? In the second part of the sentence: please be more specific about which shoeing protocols may negatively impact the hooves and why is this relevant for your article? 

Response to the reviewer:

Thank you for your suggestion. We’ve modified the content regarding the shoeing method description on page 3, lines 36-45. The context you mentioned is removed accordingly.

78: The author indicates the gap in literature but not why is it relevant to investigate differences between shoeing methods. Are there indications that one is better for the welfare of the horse or is less stressful than the other? Please spend some more explanation on this topic so that it becomes more clear why your study is relevant for the field. I assume the author is interested in psychological stress of shoeing methods and that HRV parameters are a good way to measure this but the introduction does not clearly explain this.

Response to the reviewer:

We do apologize for giving the reviewers unclear rationale in this study. We conducted this study because farriers and practitioners believe that hot shoeing provides horses with more benefits due to a subjective evaluation as it improves juxtaposition between the hoof wall and metal shoe, and horseshoes can be fitted on hooves more accurately following the hot shoeing method. However, there is no report on objective evaluation indicating whether or not horses get more benefit from hot shoeing. In this connection, the HRV was subsequently selected to quantitatively determine those benefits by comparing HRV metrics between horses receiving hot and cold shoeing protocols. Unfortunately, no report compares HRV variables between horses receiving hot and cold shoeing protocols. This rationale leads us to conduct this research, objectively evaluating the benefits of horses fitting with hot shoes. We’ve modified the introduction part of the main text to explain the study’s rationale, the gap in the literature on page 3, lines 44-45, and the objective of this study on page 5, lines 91-95.

85: this study did not include young horses, is that a deliberate choice? Were all these horse familiar with both shoeing methods previously before they were included in the study? Since the horses were recruited form equestrian clubs I assume they were habituated to wearing a girth and/or did you habituate them to wearing the polar strap?

Response to the reviewer

Since some young horses are alert and spooky during the shoeing protocol, the HRV modulation may not solely result from the effect of shoeing methods. For this reason, we decided to recruit only well-educated adult horses, easy to restrain and get accustomed to the shoeing protocol, particularly shoeing by their familiar farriers. This subject recruitment was expected to significantly reduce confounding factors and produce more accurate study results. Regarding wearing the polar strap, all horses are used to getting this strap as it is like a belt of saddle, but much smaller, fastened around the chest. So, no need to habituate them to wearing the polar strap.

93: Were these horses recruited and housed at the same stable or did you measure them at their own location? If they are in their own location and treated by their own familiar farrier, how did you control for differences in farriers?

Response to the reviewer

All horses were housed and shod in their location and treated by their familiar farrier to get rid of any confounding factors such as fear, anxiety, and other aspects that can interfere with the HRV modulation other than shoeing protocols. In this study, approximately 50 horses were mainly shod by four professional farriers who perform shoeing according to the universal concept of shoeing methods such as cutting the hoof, rasping, and nailing. Typically, there is no difference in the shoeing concept between farriers as this is not the specific therapeutic shoeing but conventional shoeing methods for healthy horses that have been adopted by farriers worldwide. A little detail of shoeing practice is observed between them; for example, some start doing the hindlegs before the front legs with respect to the horses' characteristics, etc. By doing this, the autonomic response indicated by HRV modulation can primarily reflect the impact of the shoeing protocol in this study.

105: did you soke the belt or did you wet the fur of the horses? Did you not use a lubricant gel and how about the condition of the fur did you shave them or not? All these factors do matter for how well your measurements are. What is the percentage of beat correction and did you use a correction factor?

Response to the reviewer

Before fastening the strap, the belt was soaked in the water, and the area where the strap was around the chest was also wet. In fact, horses in Thailand have relatively short hair and are often wet with sweat due to the hot and humid climate. This condition transmits a sound signal between the sensor and the Polar sports watch with no need for hair shaving and the lubricant gel on the Polar strap throughout the study period. This study analyzed RR interval data from the Polar HR monitoring device for HRV variables using the premium version of HRV analysis software (Kubios HRV Scientific; https://www.kubios.com/hrv-scientific/). This premium program (subscription program) provides artifact correction algorithms that can automatically detect misaligned and ectopic beats such as premature ventricular contractions (PVC) or other arrhythmias. This automatic correction is more accurate than manual correction in the Kubios standard version (https://doi.org/10.1080/03091902.2019.1640306). Therefore, the program automatically corrected the misaligned and ectopic beats, which we set to a medium correction level. The beat correction is allowed not greater than 5%; however, we found that the ectopic beats corrected in this study were only0.6-1.4%, according to the Kubios HRV Scientific program.

110: what do you mean by the “calm station” ?

Response to the reviewer

We mean that horses were shod in the farrier shop without loud, horrible, or unfamiliar noise during shoeing. This arrangement aims to reduce the confounding factors causing the HRV modulation other than the effect of shoeing protocols.

111: do you mean hot or cold shoeing methods by ‘protocol’ or did the farriers receive a protocol to standardize the treatment as much as possible?

Response to the reviewer

The farrier performed hot and cold shoeing by 

---

## [Decision Letter · Decision Letter 1]

31 Jan 2024

PONE-D-23-24799R1Heart rate and heart rate variability in horses receiving hot shoeing and cold shoeingPLOS ONE

Dear Dr. Chanda,

Thank you for submitting your manuscript to PLOS ONE. After careful consideration, we feel that it has merit but does not fully meet PLOS ONE’s publication criteria as it currently stands. Therefore, we invite you to submit a revised version of the manuscript that addresses the points raised during the review process.

In general the introduction was greatly improved giving a clear explanation of the scientific question aimed to be answered with the results of the current study. However, I and the  reviewers share some concerns which have to be resolved to ensure the reliability of the conslusions drawn: a main issue  to be resolved is the statistical analysis where the authors should prove that all assumptions are met for two way repeated measurement ANOVA to be performed.

The authors should answer all the questions and comment all the concernes the reviewers have risen.

We look forward to receiving your revised manuscript.

Kind regards,

Nejka Potocnik

Academic Editor

PLOS ONE

Reviewers' comments:

Reviewer's Responses to Questions

**Comments to the Author**

1. If the authors have adequately addressed your comments raised in a previous round of review and you feel that this manuscript is now acceptable for publication, you may indicate that here to bypass the “Comments to the Author” section, enter your conflict of interest statement in the “Confidential to Editor” section, and submit your "Accept" recommendation.

Reviewer #2: (No Response)

Reviewer #3: (No Response)

Reviewer #4: (No Response)

2. Is the manuscript technically sound, and do the data support the conclusions?

Reviewer #2: Partly

Reviewer #3: Partly

Reviewer #4: Yes

3. Has the statistical analysis been performed appropriately and rigorously? 

Reviewer #2: Yes

Reviewer #3: No

Reviewer #4: Yes

4. Have the authors made all data underlying the findings in their manuscript fully available?

Reviewer #2: Yes

Reviewer #3: Yes

Reviewer #4: Yes

5. Is the manuscript presented in an intelligible fashion and written in standard English?

Reviewer #2: Yes

Reviewer #3: Yes

Reviewer #4: Yes

6. Review Comments to the Author

Reviewer #2: Despite some methodological issues the article is still interesting and presents a first onset for studying the effects of shoeing on the stress response in horses. See the attached file for feedback.

Reviewer #3: Introduction

I do understand the aim of the study, however I advise authors to re-work the Introduction Section. Authors do explain why shoeing is important. However, both shoeing methods could be explained further to readers. Then, the need for this study on stress should be more detailled: why do authors expect stress when shoeing? I believe the whole paragraph on HRV methods (lines 62-90) can be shortened to one or 2 sentences. There is no need to introduce the different HRV methods in such an extend. Furthermore, looking at the explanations of authors on HRV methods, I wonder if all HRV methods were needed for this study. Also, not all HRV parameters have been validated in horses.

Material and methods

- Line 111-113: do authors mean complications after shoeing during experimental protocol? In this case, it cannot be an inclusion criteria as horses are already in the protocol. If authors mean previous shoeing, please mention. If horses were excluded after inclusion due to lameness after study shoeing, please indicate how many horses were excluded and for which reason (even if n = 0).

- Experimental design: please indicate where the horses were kept before and after shoeing, as HRV measurements depends greatly on the way horses are kept (see many studies in equine literature). The standardization of the data collection is one major effect on HRV measurements.

- Experimental design: how can authors exclude an effect of shoeing depending on the individuals? There is no consensus about HRV values in horses, and studies show largely different values depending on horses. Why did authors choice for this study design and not for example a crossover design?

- When authors should use less and the most appropriate HRV measurements, authors could use graphs instead of tables for the results, as they present a timeline in measurements.

Discussion

- Please indicate HRV measurements are repeatable in horses, both in rest (https://doi.org/10.1016/j.jevs.2016.07.006) and during activities (https://doi.org/10.1163/17552559-20220044), as it may be important to compare groups.

- I do emphasize the efforts of authors to show an effect of shoeing methods on stress in horses, and the need for more knowledge in the current need for improved welfare. However, I am still dubitative about the conclusions authors make with the acquired data. Differences may be significative but are they still related to the shoeing method and are they relevant? I would personally choice for less HRV variables, but better discussed in the whelm of significance, relevance and interpretation. The confounds and biases should me more elaborated in the discussion.

I would like to read a reviewed manuscript as I do believe this content is of importance for equine science in the context of equine welfare.

Reviewer #4: The article is interesting and well-written. However, there are the following two main points the authors should cope with to meet publication standards.

1. In the preamble of the statistical analysis section, lines 169-173, specify the levels of the two factors under consideration. In particular, how many time points there are? Remember that the two-way repeated measures ANOVA is an omnibus test statistic and cannot tell you which specific groups within each factor were significantly different from each other.

2. A fundamental part of the statistical analysis involves checking that the data can be analyzed using a two-way repeated measures ANOVA. This involves checking if your data "passes" the assumptions that are required for a two-way repeated measures ANOVA to give you a valid result. The most important assumptions to be checked are: 1) there should be no significant outliers in any combination of the related groups, 2) the distribution of the dependent variable in each combination of the related groups should be approximately normally distributed, and 3) the variances of the differences between all combinations of related groups must be equal. Just remember that if you do not run the statistical tests on these assumptions correctly, the results you get when running a two-way repeated measures ANOVA might not be valid.

7. PLOS authors have the option to publish the peer review history of their article (what does this mean?). If published, this will include your full peer review and any attached files.

Reviewer #2: No

Reviewer #3: No

Reviewer #4: No

---

## [Author Response · Author response to Decision Letter 1]

20 Feb 2024

Heart rate and heart rate variability in horses receiving hot shoeing and

cold shoeing

Dear Editor and reviewers,

I am grateful for offering to revise this manuscript. We’ve addressed all points raised by 

the reviewers and the modifications in the text were highlighted in green.

Response to the reviewer 2#

• In general, the introduction is greatly improved and clearly explained. 

Response to the reviewer

We would like to thank the reviewer’s kind words.

• I appreciate the attempt to clarify the purpose of the study. However, the issue is not resolved by changing the term ‘stress’ into ‘comfort’ in the conclusion. The point here is that in this study you are actually aiming to measure a stress response (line 46-50) and not (just) physiological activity. However, you cannot distinguish between physiological stress and psychological stress in this study, as you have not controlled for physical activity and did not record behavioural parameters. This makes interpretation of the results and differences between time periods a bit more difficult. Both psychological and physiological stress affect HRV and may be reflected in physiological parameters of stress. Therefore, the authors conclusion that hot shoed horses experience less discomfort/stress should be a bit more carefully formulated (less straightforward). Perhaps hot shoed horses spend more time standing still after the shoeing process which leads to decreased HRV parameters? This may still be an indication of discomfort, so I do not disagree with the conclusion, but it could be formulated a bit more carefully. Even more so since the differences between the methods could also be due to differences in breed/character, body condition, differences in stable size etc…..due to the setup of the study. Suggestion line 416-: ‘ANS activity responds differently in horses receiving different shoeing methods. Higher HRV 30-120 minutes after hot shoeing suggests that hot-shoed horses experience less stress compared to cold-hoed horses after shoeing. More research is needed to investigate how different shoeing techniques affect stress levels in horses.’ 

Response to the reviewer

We would like to thank the reviewer's comment and explain this study briefly. HRV is generally adopted to indicate autonomic regulation in which a reduced HRV reflect increased stress in such circumstances as pain, exercise, transportation, etc. Regarding a subjective evaluation and anecdotal evidence, the practitioner and farrier think that shoeing with hot shoes is better than cold shoeing as it improves juxtaposition between the hoof surface and metal shoe as well as burning the hoof solar surface to close the hoof tubule, leading to preventing the invasion of microorganism to the hoof wall via these tubules. However, there is no objective evidence to prove this belief. This led us to apply the HRV to indicate the autonomic regulation to objectively investigate whether or not horses undergoing hot shoeing derived more comfort than cold shoeing. In this study, we try not to use the stress response in the whole text but in the introduction to avoid misconducting since shoeing experienced horses isn’t supposed to be a stressor, as we discussed on page 16, lines 321-325.

Regarding the reviewer's comment about distinguishing between physiological stress and psychological stress, we didn’t aim to differentiate between both types of stress but rather the overall autonomic response following different shoeing types. However, we designed the study to minimise psychological stress by 1) experimenting on horses that are accustomed to fitting with each type of farrier practice and 2) shoeing horses by familiar farriers and places. By doing this, the changes in HRV variables in this study were expected to result mainly from different physiological stimuli following hot and cold shoeing. We’ve added this document in the material and methods section on page 5, lines 104-106. As measured before shoeing, HRV variables didn’t differ between HSH and CSH groups, implying that horses in both groups showed similar autonomic responses regardless of the type of physical activity, sex, breed, etc., before shoeing (as shown in tables 1-3). We’ve added this information in the discussion part on pages 14-15, lines 283-287. Of note, the behavioral parameter is not practical in this study because conventional shoeing in healthy horses carried out by familiar farriers may not trigger behavioral changes during the shoeing period, and it might not be suitable for assessing the autonomic response to shoeing methods in horses (as stated on page 3, lines 45-49). That’s why we haven’t implemented the behavioral measurement in this study.

A major difference between both shoeing types is that heating the metal shoe, searing the hoof solar surface and adjusting the shoes to fit the solar surface were performed in hot shoeing. On the other hand, the metal shoes were directly adjusted to fit the solar surface without heating in cold shoeing. In fact, heating shoes aids in metal shoe adjustment in hot shoeing and takes less time than adjusting non-heating shoes in cold shoeing. So, the time taken for horseshoeing was somewhat similar in both groups, and horses fitting with hot shoes didn’t take longer to stand still than horses fitting with cold shoes. Each horse was walked to their stable after completing the shoeing protocol. This is due to the fact that overall HRV in this study was either constant throughout the study or increased during and after shoeing. A decrease in “some HRV variables, such as stress index and SNS index” indicated a positive impact of such shoeing.

Considering the HRV modulation during shoeing, there were almost no changes in HRV variables (except stress index, which was reduced during shoeing) during shoeing but increased at around 90-120 minutes after shoeing in the CSH group. These results suggested that CSH horses were not under stress conditions during and after shoeing. In contrast, various HRV variables increased in HSH horses during shoeing, extending to 120 minutes after shoeing. Moreover, multiple HRV variables were higher in HSH than CSH horses at given time points after shoeing. An increase in HRV reflected more relaxation or comfort in HSH horses. That’s why we avoid using the terms less stress, reduced stress, or minimal stress in horses receiving shoeing but apply the term “more comfort” following an increase in HRV in horses receiving hot shoeing. This result implies that HSH horses not only showed no stress conditions during shoeing but also derived more comfort after shoeing and demonstrated more comfort than CSH horses after shoeing.

• In this study the shoeing process itself does not lead to increased HRV parameters compared to the pre-period. This might indicate that horses do not experience psychological stress from either procedure. This is in contrast to what I would expect, but could be due to the fact that horses are accustomed to the process already and because you selected them for easy shoeing (creating a bias). Therefore mention this in your M&M section because it puts your results into perspective. If you would change the inclusion criteria you might find different results as some horses would indeed experience shoeing as a stressor. However, it also depends on how the pre measurement is taken, if this is already done within a restraint box (or a separate place where the shoeing takes place) the horses might already anticipate to the procedure and experience psychological stress. This needs to be addressed in the discussion. It would have been preferable if the pre measurement had been done in the stable so the conditions of pre and post would be comparable. 

Response to the reviewer

We would like to thank the reviewer's comment and explain a little bit about the study protocol. In this study, the psychological stress was expected to be eliminated by the study design, as stated on page 5, lines 104-106. we confirm that we didn’t aim to create bias but designed the experiment to “minimize psychological stress” by 1) experimenting on horses that are accustomed to fitting with each type of farrier practice and 2) shoeing horses by familiar farriers and places. If we’ve selected horses who aren’t accustomed to the farrier practice (new horses or horses that haven’t been shod before, for example) or have designed the experiment as the cross-over design (one horse that is only familiar with either hot or cold shoeing receives both shoeing protocols), we could have created “the predictable stressor”, leading to decreased HRV and increased stress due to fear and anxiety in those horses and the results cannot reflect the effect of shoeing but the fear or anxiety on autonomic modulation. However, to give more precise information, we’ve added the document on why we selected the experience horses for shoeing in this study in the material and method part on page 5, lines 95-96 and 104-106.

In the present study, the pre-shoeing measurement was done in designated stables within the farrier place for 30 minutes before leading them to be shod in the calm station in front of the designated stables (figure 1) or next to the designated stable (figure 2). The waiting stable is similar to the horse stable in terms of stable type, design, size, and straw bedding. Horses were allowed to have hay and drink water as if they were housed in their stable before shoeing. For this reason, HRV values at given time points and pre-shoeing values could be comparable. This message has been added to provide more precise information on page 6, lines 114-117.

 Supposing that shoeing for experienced horses is a non-stress condition. Still, it should be “a reliever “, as the shoeing method is conducted to correct hoof discomfort during the late period of 4-6 weeks of scheduled shoeing. Since the hoof wall grows over time, the increased weight-bearing length following toe growth and a decreased heel angle may cause hoof angle displacement. Moreover, metal shoes do not change the size, and the maximum expansion of the hoof wall is limited. These events cause gradual discomfort in the shod feet of horses, as we discussed on page 16, lines 321-324. 

Not surprisingly, the shoeing protocol is supposed to relieve discomfort, as indicated by reduced stress index and sympathetic nervous system index in horses during shoeing with both protocols. More importantly, the differences in HRV modulation between shoeing types were observed in horses after shoeing, reflecting that autonomic response differed between horses receiving distinct farrier practices. If shoeing had been the stressor in experienced horses, the HRV could have been reduced (not increased or remained constant), indicating increased stress during shoeing. 

• Remark 85: mention in your results the selection process of the horses and that they are accustomed to wearing a girth.

Response to the reviewer

We would like to thank the reviewer's comment. We’ve added the information regarding the criteria for horse selection in this study on page 5, lines 95-99 and lines 104-106. As per our routine protocol, the HRM device was equipped on horses for at least 10 minutes to get them accustomed to the setting before the recording started. We’ve added this sentence on page 6, lines 114-117.

• Remark 93: I assume that the farriers work in certain stables and also perform either hot or cold shoeing so you did not control for this? This also means you cannot distinguish the effect of the shoeing procedures from the farrier and/or location effects so this makes it difficult to interpret the results. Please also explain this in detail in your M&M.

Response to the reviewer

We would like to thank the reviewer's comment. Suppose we organised the controlled shoeing protocol by assigning only one farrier to shoe all horses or arranged them to shoe horses in equal numbers and shoeing protocols (regardless of familiarity). In that case, many horses may have experienced anxiety, fear or uncertainty, causing distortion in HRV modulation due to anxiety and fear rather than the shoeing protocol. Similarly, staying all experiment horses in the same designated places before, during and after shoeing may have caused anxiety and fear, predisposing them to stress from unfamiliar locations and distorting the HRV modifications.

As stated earlier in this response to the reviewer and within the text on page 5, lines 95-96 and 104-106, each horse received the shoeing protocol by familiar farrier, method and place. In fact, both shoeing protocols were conducted virtually all the same unless the seared solar surface was before adjustment and nailing in hot shoeing and nailing directly after adjustment in cold shoeing. Moreover, shoeing, in reality, causes no pain as all cutting, floating and nailing were done on the insensitive hoof wall by professional farriers, as it is evident that no HRV variables were reduced during shoeing. In addition, the condition of waiting stable for shoeing was similar to their stable (stable type, size and bedding type). So, the arrangement in this study aimed to avoid or minimise the “psychological factor” from shoeing by unfamiliar farriers and locations during the shoeing. Thus, changes in HRV variables were expected to result from horses' physiology (discomfort, remaining neutral or more comfort) during and after shoeing in horses. We’ve also discussed this issue on pages 14-15, lines 283-290.

• Remark 105+110+111 mention this in the M&M section

Response to the reviewer

We do apologize that we didn’t catch this reviewer's comment. As lines 105, 110 and 111 are in the material and methods section. 

• Remark 191: See also previous remarks. The horse might be simply more comfortable in the stable compared to the ‘calm box’ and this does not necessarily have to be caused by the shoeing process. If you want to talk about stress/comfort during the post period you should be able to compare this to a pre period in which the horse is within the same conditions (so its own stable). I really think that the term ‘calm box’ is misleading, if this is the station where the horses are normally shoed than this location might actually be associated with stress. This might explain the increase in HRV in the post period compared to the pre period but not the fact that this difference was not found in the cold-shoeing method. However, the differences between the two shoeing methods during the post period may also be caused by differences in location (for example bedding) due to the setup of the study. This actually presents two methodological problems that needs to be addressed in the discussion at least. Actually location and farrier should have been controlled for or included as a factor in your model. This cannot be changed anymore but it should be considered when formulating the conclusions and mentioned in the discussion.

Response to the reviewer

We would like to thank the reviewer's comment. In this study, the HRV metrics were measured and computed 30 minutes pre-shoeing at designated stables where the conditions were similar to their stable, as stated earlier in this reviewer’s responses. Horses have shod in calm stations in front of the stable (as shown in Figure 1) or next to the designated stable (as shown in figure 2). In fact, all horses were usually housed in a similar standard box size (3x5 m2) with straw bedding. The designated stable for horses waiting for shoeing is similar to the housing stable, including bedding. As the reviewer gave us a comment that “The horse might be simply more comfortable in the stable compared to the ‘calm box’, and this does not necessarily have to be caused by the shoeing process”, it would have been true if we found an increase in HRV after shoeing (when they were in their own stables) “in horse receiving both shoeing types.” In fact, the increased HRV was found mainly in hot-shoed horses even though all conditions of the housing stable for horses in both shoeing types were somewhat similar, including straw bedding type. This result could reflect different autonomic responses following physiological factors in horses with different shoeing protocols. We’ve added the details in the material and method that the study was designed to avoid or minimise the psychological stress

---

## [Decision Letter · Decision Letter 2]

11 Mar 2024

PONE-D-23-24799R2Heart rate and heart rate variability in horses undergoing hot and cold shoeingPLOS ONE

Dear Dr. Chanda,

Thank you for submitting your manuscript to PLOS ONE. After careful consideration, we feel that it has merit but does not fully meet PLOS ONE’s publication criteria as it currently stands. Therefore, we invite you to submit a revised version of the manuscript that addresses the points raised during the review process.

In the revised manuscript some reviewers' comments have not been addressed, esspetially the satisfactory answers to the reviewer 2 are missing.  Please see the uploaded file of the reviewer 2 provide a feedback.

We look forward to receiving your revised manuscript.

Kind regards,

Nejka Potocnik

Academic Editor

PLOS ONE

Journal Requirements:

Reviewers' comments:

Reviewer's Responses to Questions

**Comments to the Author**

1. If the authors have adequately addressed your comments raised in a previous round of review and you feel that this manuscript is now acceptable for publication, you may indicate that here to bypass the “Comments to the Author” section, enter your conflict of interest statement in the “Confidential to Editor” section, and submit your "Accept" recommendation.

Reviewer #2: (No Response)

Reviewer #3: All comments have been addressed

Reviewer #4: All comments have been addressed

2. Is the manuscript technically sound, and do the data support the conclusions?

Reviewer #2: No

Reviewer #3: Yes

Reviewer #4: Yes

3. Has the statistical analysis been performed appropriately and rigorously? 

Reviewer #2: Yes

Reviewer #3: Yes

Reviewer #4: Yes

4. Have the authors made all data underlying the findings in their manuscript fully available?

Reviewer #2: Yes

Reviewer #3: Yes

Reviewer #4: Yes

5. Is the manuscript presented in an intelligible fashion and written in standard English?

Reviewer #2: Yes

Reviewer #3: Yes

Reviewer #4: Yes

6. Review Comments to the Author

Reviewer #2: The authors have not addressed the comments on the previous version to my satisfaction. Furthermore the discussion only raised the same issues even more with the current changes. Please see the uploaded file for feedback.

Reviewer #3: Thank you to the authors for the revised manuscript. The revised manuscript encompasses answers to the comments, and the improved version meets in my opinion the requirements for publication. When a revised version is re-submitted, I should advise authors to discuss more the effects/advantages/disadvantages of both shoeing techniques on locomotor apparatus. And I should be curious to read the thoughts of authors about the reason why hot shoeing induces higher HRV parameters. This is however a personal preference, to offer the readers with discussion items.

Reviewer #4: The authors did a good job in revising the paper according to the received suggestions. I suggest accepting the paper in its current form.

7. PLOS authors have the option to publish the peer review history of their article (what does this mean?). If published, this will include your full peer review and any attached files.

Reviewer #2: No

Reviewer #3: No

Reviewer #4: No

---

## [Author Response · Author response to Decision Letter 2]

19 Mar 2024

Heart rate and heart rate variability in horses undergoing hot and cold shoeing

Dear reviewer

We’d like to thank the reviewer for giving valuable comments on this manuscript. The corrections are highlighted in green within the text.

Reviewer’s comments

Line 32 (abstract): The conclusion that horses experience more comfort cannot be drawn. Please only state that differences in HRV were found indicating increased sympathovagal activity in horses that were hot shoed compared to cold shoed horses.

Response to the reviewer

We’ve modified the statement according to the reviewer’s comment on page 2, lines 31-32.

Line 45-49: Behavioural parameters can be used to assess stress. This is possible during shoeing practices, for example through recording facial expressions (Ask et al. 2020, van Loon et al. 2015) but also stress related behaviours (Visser et al. 2009, Young et al. 2012). Recording behaviours may have provided more insight into differences in activity. That for this study you decided not to record this for (practical?) reasons is understandable but does not make behaviour an unsuitable parameter. Please move this section to the discussion and add it as a limitation to the study. 

Response to the reviewer

We’ve removed this context from the introduction part and added it as the limitation in the discussion part on pages 19-20, lines 409-411.

Line 143-144: Please indicate which level of noise detection you used and what levels of beat correction were considered acceptable (were no recondings excluded because the recording was to noisy?).

Response to the reviewer

We’ve added the information regarding the level of noise detection and artifact correction on page 7, lines 142-144.

Line 283-286: Please move these lines to the M&M section. The changes in HRV variable cannot be expected to result mainly from physiological factor of the shoeing process as you did not control for differences in location and/or locomotion in the horses. Confounding factors cannot be excluded, for example horses of the cold shoeing location could be anticipating to feeding during your post shoeing measurement explaining the higher HRV compared to hot shoed horses (your M&M section does not give information on differences in feeding times and daily routines). The lack of differences in autonomic responses before the study commenced only eliminates the possibility of individual differences influencing these results. 

Response to the reviewer

We’ve removed this context from the discussion part and added it in the materials and methods on pages 5, lines 99-100.

Line 331-333: Here you use the term stress and not comfort which is confusing for the reader. Furthermore, the conclusion cannot be drawn from your results as your baseline measurement is not carried out under the right circumstances. There is limited evidence that the shoeing process provides comfort and you do not substantiate the statement that gradual discomfort may arise in shod feet. In addition, you do not consider that the process of shoeing might also be stressful. As stated before horses are measured not in their own stable during baseline conditions, placing them in a different stable might already be a stressor, even more so since they may be anticipating to be shod. The only measurement that is done in their own stable is the post shoeing measurement. This may explain the increase in vagal tone as well and does not necessarily have to be contributed to the comfort of the new shoes. This should be discussed and mentioned as a limitation.

Response to the reviewer

We’ve modified the statement to use more comfort instead of reduced stress on page 16, line 323. We’ve also stated that the measurement of HRV not in its own stable may be considered a stressor in the limitation section of the discussion part on page 19, lines 406-409.

Line 397-409: The subjective reporting of more benefits of hot-shoeing compared to cold shoeing is not really something that you can use as a starting point of this alinea. The first sentence can be used as an explanation but not as a fact, so it is better to begin with your own findings (line 399-400) and then state that this is in line with subjective evidence that seems to indicate that hot shoeing is more beneficial that cold shoeing. In this way you also avoid drawing hard conclusions on this subject which cannot be validated by your own research. As stated before confounding factors may contribute to the differences between hot and cold shoed horses as well, so the statement that this study provides objective evidence for greater benefits of hot shoeing should be done more carefully. Especially since you also state in lines 404-407 adverse effects and risks of lameness through shoeing practices you seem to contradict the previous assumption that (hot) shoeing should lead to more comfort.

Response to the reviewer

We’ve modified the statement according to the reviewer's suggestion on page 19, lines 391-394. Regarding the last sentence of this paragraph, we’d like to convince the reader that although we show that horses demonstrated increased HRV, reflecting increased vagal tone, since 30 minutes post-shoeing in hot-shoed horses and the increase was found later at 120 minutes post-shoeing in cold shooed horses. This finding may assume that conventional shoeing leads to more comfort in horses. However, as stated in this part, metal shoeing is not a natural manner that may cause adverse effects in long-term shoeing in horses. So, the evaluation of gait analysis using HRV determination together with fundamental lameness examination in shod horses during locomotion and exercise needs to be further investigated.

Line 410: Start this alinea with the statement that a limitation of the study was that a cross-over design was not used and then explain why this is not possible.

Response to the reviewer

We’ve modified the statement in the limitation section on page 19, lines 403-405.

Line 420: The line ‘Conventional shoeing protocols provide horses comfort during and after shoeling’ is a too strong conclusion that cannot be drawn due to the setup of your study. Suggestion: ‘ANS activity responds differently in horses undergoing different shoeing methods, with greater HRV 30-120 minutes after hot shoeing compared to cold shoeing. More research is needed to substantiate whether these differences can be attributed to the fact that horses experience more comfort from hot shoeing compared to cold shoeing.’ 

Response to the reviewer

We’d like to thank the reviewer for the comment and provide an appropriate statement for this conclusion. We’ve adopted this context and added it in the conclusion on page 20, lines 416-419.

---

## [Decision Letter · Decision Letter 3]

16 Apr 2024

PONE-D-23-24799R3Heart rate and heart rate variability in horses undergoing hot and cold shoeingPLOS ONE

Dear Dr. Chanda,

Thank you for submitting your manuscript to PLOS ONE. After careful consideration, we feel that it has merit but does not fully meet PLOS ONE’s publication criteria as it currently stands. Therefore, we invite you to submit a revised version of the manuscript that addresses the points raised during the review process.

In the new version pf the manuscript, all the issues were addressed and the manuscript has improved greatly. A few points in the discussion still need attention. Please correct!

We look forward to receiving your revised manuscript.

Kind regards,

Nejka Potocnik

Academic Editor

PLOS ONE

Journal Requirements:

Reviewers' comments:

Reviewer's Responses to Questions

**Comments to the Author**

1. If the authors have adequately addressed your comments raised in a previous round of review and you feel that this manuscript is now acceptable for publication, you may indicate that here to bypass the “Comments to the Author” section, enter your conflict of interest statement in the “Confidential to Editor” section, and submit your "Accept" recommendation.

Reviewer #2: (No Response)

2. Is the manuscript technically sound, and do the data support the conclusions?

Reviewer #2: Yes

3. Has the statistical analysis been performed appropriately and rigorously? 

Reviewer #2: Yes

4. Have the authors made all data underlying the findings in their manuscript fully available?

Reviewer #2: Yes

5. Is the manuscript presented in an intelligible fashion and written in standard English?

Reviewer #2: Yes

6. Review Comments to the Author

Reviewer #2: The authors have addressed the points made previously and the manuscript has improved greatly. A few points in the discussion still need attention. If these are addressed correctly I do feel that this paper may be published.

Line 299-314: This section addresses the methodology and seems more appropriate within the M&M section. If the authors feel that this should remain within the discussion, please move the section more to the beginning (after line 282) and make clear why it is relevant to discuss this point. For example explain whether there is a debate within literature how HRV should be measured reliably and start the alinea with explaining this.

Line 328-333: This section is not a discussion but a repetition of the results, as you do not relate to the existing literature. As the discussion section is already quite elaborate and these points are not the main ones, my suggestion would be to just leave it out.

Line 391-401: The first sentence in still too bold, change it into: 'The findings of the study seem to confirm subjective evidence...." for example. Then still this sentence is more a conclusion from the previous section. It does not make clear what the rest of the alinea is about. So start with the main point of this alinea, what would you like to discuss here? The relationship of HRV and lameness? Perhaps here you could relate to the point that measuring locomotion and/or behavioural parameters may provide relevant/additional information?

Line 403-414: Generally I do appreciate that an alinea is added that discusses the limitations, but sometimes a bit more explanation is needed. For example line 406-408: The sentence is not formulated correctly and it is unclear, it starts with Since....but since what? In addtition line 411 suffers from the same issue, explain why/how including behavioural parameters may render relevant or addition information that may confirm stress for example and clarify HRV differences. LIne 412-414, explain why inter-individual differences is an issue. Probably you mean that this may cause a bias as there may be influences of the locations on differences between hot/cold shoed groups?

7. PLOS authors have the option to publish the peer review history of their article (what does this mean?). If published, this will include your full peer review and any attached files.

Reviewer #2: No

---

## [Author Response · Author response to Decision Letter 3]

17 Apr 2024

Reviewer #2: 

The authors have addressed the points made previously and the manuscript has improved greatly. A few points in the discussion still need attention. If these are addressed correctly I do feel that this paper may be published.

Line 299-314: This section addresses the methodology and seems more appropriate within the M&M section. If the authors feel that this should remain within the discussion, please move the section more to the beginning (after line 282) and make clear why it is relevant to discuss this point. For example explain whether there is a debate within literature how HRV should be measured reliably and start the alinea with explaining this.

Response to the reviewers

We’d like to thank the reviewer for this valuable suggestion. After careful reading and interpretation again, we decided to remove this section from the discussion part as it seemed to be redundant. The core context in this discussion part has already been mentioned in the M&M part, for example, the validation of using HRM devices in horses (page 5, lines 102-103). The application of automatic artefact and noise correction provided by the Kubios premium program (page 7, lines 135-145).

Line 328-333: This section is not a discussion but a repetition of the results, as you do not relate to the existing literature. As the discussion section is already quite elaborate and these points are not the main ones, my suggestion would be to just leave it out.

Response to the reviewers

We’ve removed this section from the discussion section accordingly.

Line 391-401: The first sentence in still too bold, change it into: 'The findings of the study seem to confirm subjective evidence...." for example. Then still this sentence is more a conclusion from the previous section. It does not make clear what the rest of the alinea is about. So start with the main point of this alinea, what would you like to discuss here? The relationship of HRV and lameness? Perhaps here you could relate to the point that measuring locomotion and/or behavioural parameters may provide relevant/additional information?

Response to the reviewers

We’d like to thank the reviewer for the suggestion. We’ve revised this section to address the study’s findings and further study of HRV modulation in horses during locomotion and exercise in athletic horses on page 18, lines 371-377.

Line 403-414: Generally I do appreciate that an alinea is added that discusses the limitations, but sometimes a bit more explanation is needed. For example line 406-408: The sentence is not formulated correctly and it is unclear, it starts with Since....but since what? In addtition line 411 suffers from the same issue, explain why/how including behavioural parameters may render relevant or addition information that may confirm stress for example and clarify HRV differences. LIne 412-414, explain why inter-individual differences is an issue. Probably you mean that this may cause a bias as there may be influences of the locations on differences between hot/cold shoed groups?

Response to the reviewers

We’d like to thank the reviewer for the suggestion. We’ve revised the limitation of the study section to address the reviewer's concern on page 18-19, lines 378-390.

---

## [Decision Letter · Decision Letter 4]

23 May 2024

Heart rate and heart rate variability in horses undergoing hot and cold shoeing

PONE-D-23-24799R4

Dear Dr. Chanda,

We’re pleased to inform you that your manuscript has been judged scientifically suitable for publication and will be formally accepted for publication once it meets all outstanding technical requirements.

Kind regards,

Nejka Potocnik

Academic Editor

PLOS ONE

Additional Editor Comments (optional):

Reviewers' comments:

Reviewer's Responses to Questions

**Comments to the Author**

1. If the authors have adequately addressed your comments raised in a previous round of review and you feel that this manuscript is now acceptable for publication, you may indicate that here to bypass the “Comments to the Author” section, enter your conflict of interest statement in the “Confidential to Editor” section, and submit your "Accept" recommendation.

Reviewer #2: All comments have been addressed

2. Is the manuscript technically sound, and do the data support the conclusions?

Reviewer #2: Yes

3. Has the statistical analysis been performed appropriately and rigorously? 

Reviewer #2: Yes

4. Have the authors made all data underlying the findings in their manuscript fully available?

Reviewer #2: Yes

5. Is the manuscript presented in an intelligible fashion and written in standard English?

Reviewer #2: Yes

6. Review Comments to the Author

Reviewer #2: All comments have been addressed, the limitations of the study have been elaborated on in the discussion section. Use of the english language is satisfactory. Details on statistical analyses have been provided and explained.

7. PLOS authors have the option to publish the peer review history of their article (what does this mean?). If published, this will include your full peer review and any attached files.

Reviewer #2: No

---

## [Editor Report · Acceptance letter]

29 May 2024

PONE-D-23-24799R4 

PLOS ONE

Dear Dr. Chanda, 

I'm pleased to inform you that your manuscript has been deemed suitable for publication in PLOS ONE. Congratulations! Your manuscript is now being handed over to our production team.

Kind regards, 

on behalf of

Dr. Nejka Potocnik 

Academic Editor

PLOS ONE